

# Biogeography, diversity and environmental relationships of shelf and deep-sea benthic Amphipoda around Iceland

Anne-Nina Lörz[1], Stefanie Kaiser[2], Jens Oldeland[3], Caroline Stolter[4], Karlotta Kürzel[5] and Saskia Brix[6]

[1] Institute for Marine Ecosystems and Fisheries Science, Universität Hamburg, Hamburg, Germany
[2] Faculty of Biology and Environmental Protection, Department of Invertebrate Zoology and Hydrobiology, University of Łódź, Lodz, Poland
[3] Eco-Systems, Hamburg, Germany
[4] Department Biology, Zoological Institute, Universität Hamburg, Hamburg, Germany
[5] Department Biology, Universität Hamburg, Hamburg, Germany
[6] Deutsches Zentrum für Marine Biodiversität, Senckenberg Nature Research Society, Hamburg, Germany

Corresponding author
Anne-Nina Lörz, Anne-Nina.
Loerz@uni-hamburg.de

## ABSTRACT

The waters around Iceland, bounding the Northern North Atlantic and the Nordic seas, are a region characterized by complex hydrography and seabed topography. This and the presence of the Greenland-Iceland-Faroe-Scotland ridge (GIFR) are likely to have a major impact on the diversity and distribution of the benthic fauna there. Biodiversity in this region is also under increasing threat from climate-induced changes, ocean warming and acidification in particular, affecting the marine realm. The aim of the present study was to investigate the biodiversity and distributional patterns of amphipod crustaceans in Icelandic waters and how it relates to environmental variables and depth. A comprehensive data set from the literature and recent expeditions was compiled constituting distributional records for 355 amphipod species across a major depth gradient (18–3,700 m). Using a 1° hexagonal grid to map amphipod distributions and a set of environmental factors (depth, pH, phytobiomass, velocity, dissolved oxygen, dissolved iron, salinity and temperature) we could identify four distinct amphipod assemblages: A Deep-North, Deep-South, and a Coastal cluster as well as one restricted to the GIFR. In addition to depth, salinity and temperature were the main parameters that determined the distribution of amphipods. Diversity differed greatly between the depth clusters and was significantly higher in coastal and GIFR assemblages compared to the deep-sea clusters north and south of the GIFR. A variety of factors and processes are likely to be responsible for the perceived biodiversity patterns, which, however, appear to vary according to region and depth. Low diversity of amphipod communities in the Nordic basins can be interpreted as a reflection of the prevailing harsh environmental conditions in combination with a barrier effect of the GIFR. By contrast, low diversity of the deep North Atlantic assemblages might be linked to the variable nature of the oceanographic environment in the region over multiple spatio-temporal scales. Overall, our study highlights the importance of amphipods as a constituent part of Icelandic benthos. The strong responses of amphipod communities to certain water

mass variables raise the question of whether and how their distribution will change due
to climate alteration, which should be a focus of future studies.

# INTRODUCTION

Human impacts on the world's oceans are fundamentally altering the biogeography and biodiversity of marine communities (*Lotze et al., 2006*; *Halpern et al., 2008*). Cumulating effects of climate change, resource exploitation and pollution are particularly pronounced in the Northern Hemisphere, and some of these changes have already evoked significant biotic responses, such as shifts in distribution and abundance (*e.g.*, *Harley et al., 2006*; *Jones et al., 2014*; *Birchenough et al., 2015*; *Hiddink, Burrows & García Molinos, 2015*). The pace and strength of global warming and increased atmospheric $CO_2$ may be faster and greater in the ocean than in the terrestrial realm (*Burrows et al., 2011*), but our knowledge of the consequences for the marine biota is limited (*Richardson & Poloczanska, 2008*). Uncovering distribution patterns of species and the identification of the ecological and evolutionary factors and processes responsible for them is therefore vital for predicting biodiversity responses to global change.

A complex array of mechanisms have been identified to determine the distribution of species on multiple spatial and temporal scales (*Leibold et al., 2004*). Abiotic variables confine the space that species occupy according to their physiological limits (*Chase & Leibold, 2003*). Species' dispersal capacity alongside their evolutionary heritage defines the size of their realized distribution (*Grantham, Eckert & Shanks, 2003*; *Hilário et al., 2015*; *Baco et al., 2016*). Finally, biological relationships are known to structure spatial patterns of species in many ways, such as those associated with competitors, consumers, and facilitators (*Jablonski, 2008*; *Bascompte, 2009*).

Environmental differences may be less obvious in the deep sea (>200 m) than in the shallows. It is now clear, though, that there is considerable spatial and temporal variation in the physical and biological properties to which species are exposed and which determine their distribution. Processes associated with sediment properties, temperature, salinity, nutrient input and dissolved oxygen are among the main drivers for structuring biodiversity and its geographical distribution (*Levin et al., 2001*; *Schnurr et al., 2018*). However, there is still a lack of understanding of distribution boundaries in the marine realm and even less so in the deep sea (*Lourie & Vincent, 2004*; *Rex et al., 2005*), making it difficult to predict how communities will respond in the wake of a changing ocean.

The waters around Iceland and adjoining seas represent a spatially heterogeneous environment with steep gradients that promote distinct habitats and related communities. As a boundary region between temperate North Atlantic, and polar waters, they are also considered to be very susceptible to climatic changes (*Astthorsson, Gislason & Jonsson, 2007*; *Eiríksson et al., 2011*). Iceland is located on top of the mid-Atlantic ridge and is

criss-crossed by several topographic barriers that determine the flow of water masses and ultimately the distribution of species. At the forefront is the Greenland-Iceland-Faroe ridge (GIFR), which stretches from Scotland and the Faroes *via* Iceland to Greenland, and and restricts the exchange of water masses between the warm, salty North Atlantic waters and the cold and less salty Nordic Seas (*Hansen et al., 2008*). As a result, seabed temperature and salinity differ strongly between areas north and south of the GIFR, which, in turn, can lead to marked differences in species compositions (*Weisshappel & Svavarsson, 1998*; *Weisshappel, 2000*; *Bett, 2001*; *Weisshappel, 2001*; *Brix & Svavarsson, 2010*; *Dauvin et al., 2012*; *Jochumsen, Schnurr & Quadfasel, 2016*; *Schnurr et al., 2018*). Alterations of the physicochemical environment, including temperature rise, ocean acidification, and salinity, have already been observed around Iceland (*Astthorsson, Gislason & Jonsson, 2007*; *Olafsson et al., 2009*; *Seidov et al., 2015*; *Jochumsen, Schnurr & Quadfasel, 2016*). Knowledge on the most important environmental parameters structuring deep-sea benthic communities would therefore allow prediction of future changes for those communities.

Amphipod crustaceans are very common and diverse across marine benthic habitats (*Just, 1980*; *De Broyer & Jazdzewski, 1996*; *Lörz, 2010*; *Stransky & Brandt, 2010*; *Brix et al., 2018*; *Jażdżewska et al., 2018*), and also in Icelandic waters (*Weisshappel, 2000*; *Weisshappel, 2001*; *Dauvin et al., 2012*; *Brix et al., 2018*). Their occurrence in a wide variety of marine environments, in turn translates into a diverse feeding types that comprise detritivores, suspension-feeders, predators, and scavengers amongst others (*Guerra-García et al., 2014*). But they also play a central role in the marine food web (*e.g.*, *Lörz, 2010*; *Nyssen et al., 2002*). Amphipods, as a member of the crustacean superorder Peracarida, have a brooding life style, from which a limited dispersal capacity is derived for most species making them potentially very susceptible to environmental change (*e.g.*, *Jablonski & Roy, 2003*; but see *Lucey et al., 2015*). Exceptions are purely pelagic species (*e.g.*, within the Hyperiidea) or species of the highly mobile scavenging guild.

The aim of this study was to identify the main factors influencing the distribution and biodiversity of marine amphipods in the waters around Iceland. This could provide hints as to which variables could most importantly affect distribution changes as a result of climate change. For this purpose, a comprehensive data—set from the literature and recent expeditions was compiled constituting distributional records for 355 species across a major depth gradient (18–3,700 m). These come from historical missions, in particular the Danish Ingolf expedition (1895 and 1896), which carried out sampling in Icelandic and West Greenlandic waters (*Stephensen, 1944b*), but mainly from sampling as part of BIOICE (Benthic Invertebrates of Icelandic Waters) and IceAGE (Icelandic marine Animals: Genetics and Ecology) projects (*e.g.*, *Brix et al., 2014*). Earlier community analyses of the Icelandic amphipod fauna identified depth as a strong driver of species distributions, but water mass properties were also important (*Dauvin et al., 2012*; *Brix et al., 2018*). In this regard, the GIFR appears to act as a major, albeit surmountable distributional barrier (*Weisshappel & Svavarsson, 1998*; *Weisshappel, 2000*; *Weisshappel, 2001*; *Dauvin et al., 2012*; *Brix et al., 2018*). Therefore, we tested whether geographical

distinctions of Iceland, mainly determined by the GIFR and depth, are mirrored by the distribution of benthic amphipods.

## MATERIALS & METHODS

### Amphipoda data

We compiled data on occurrences and abundance of 355 amphipod species for 532 localities from the literature. The following expeditions and respective data sources were used: extensive literature search, data from BIOICE and IceAGE expeditions. The assembled dataset was highly heterogenous regarding sampling effort and method, time, location and date of the different expeditions. Many only listed one or two species, in particular the historic data from *e.g.*, *Boeck (1861)*, *Hansen (1887)* and *Stephensen (1933*, *1938*, *1942*, *1944a*, *1944b)* only providing occurrence data. However, other had high abundances of individuals (max: 2,709) and high species richness (max: 72). Due to this high heterogeneity, we aggregated the data at a coarser spatial resolution.

A common approach is to construct a coarse rectangular grid in which species occurrences are joined. We constructed a hexagonal grid using QGIS (*QGIS Development Team, 2019*) with a horizontal diameter of 1° per grid cell. Within each grid cell, the occurrence and abundance information were pooled, so that a grid cell contained information from multiple localities but species were not double counted, yet the sum of the abundances per species could be calculated. Hexagonal grids have several advantages over rectangular grids, *e.g.*, symmetric neighbourhood relations or reduced edge effects (*Birch, Oom & Beecham, 2007*). For our study the most compelling reason to favour a hexagonal grid was the match of the polygons to the coastlines of Iceland and Greenland. Hexagonal grids provided a much better fit to this jagged pattern with an appropriate size, whereas rectangular grid cells would have to be much smaller and would then be too small for the purposes of our sampling. Given the case that many of our samples were near the coast, the hexagonal design clearly improved our sampling design.

### Environmental layers

We extracted twelve variables from the Bio-Oracle 2.0 database (*Assis et al., 2018*) using the *sdmpredictors* package (*Bosch, Tyberghein & De Clerck, 2018*). Variables were chosen to represent major environmental deep-sea gradients (Table 1). All variables, except minimum depth, represented long-term maximum values modelled at minimum depths on a raster with 7 km$^2$ resolution per cell. In order to use the parameters on the same spatial scale as the species data, we aggregated the raster data to the scale of the hexagonal grid cells by calculating the mean raster value for each grid cell. Hexagons then represented the summed species abundances and averaged environmental data.

We analysed the environmental data for multicollinearity on the level of the hexagons. We calculated a Pearson correlation matrix (AppS1) for all environmental layers and removed all layers with a Pearson's r above 0.75. As expected, we found strong correlation between parameters of the same information type, *i.e.*, Chl-*a* and primary productivity or all nutrient related parameters. Finally, we retained the following parameters: depth, pH, phytobiomass, velocity, dissolved oxygen, dissolved iron, salinity and temperature.

**Table 1 Environmental parameters.**

| Acronym | Parameter | Units | Source |
|---------|-----------|-------|--------|
| depth | Bathymetry | meters | GEBCO URL: http://gebco.net Bathymetry URL: http://www.emodnet-bathymetry.eu/ |
| chla | Chlorophyll concentration | mg/m$^2$ | Global Ocean Biogeochemistry NON ASSIMILATIVE Hindcast (PISCES) URL: http://marine.copernicus.eu/ |
| vel | Current velocity | m/s | Global Ocean Physics Reanalysis ECMWF ORAP5.0 (1979–2013) URL: http://marine.copernicus.eu/ |
| dO$_2$ | Dissolved oxygen concentration | μmol/m$^2$ | Global Ocean Biogeochemistry NON ASSIMILATIVE Hindcast (PISCES) URL: http://marine.copernicus.eu/ |
| dFe | Dissolved iron concentration | μmol/m$^2$ | Global Ocean Biogeochemistry NON ASSIMILATIVE Hindcast (PISCES) URL: http://marine.copernicus.eu/ |
| dP | Phosphate concentration | μmol/m$^2$ | Global Ocean Biogeochemistry NON ASSIMILATIVE Hindcast (PISCES) URL: http://marine.copernicus.eu/ |
| dNO$_3$ | Nitrate concentration | μmol/m$^2$ | Global Ocean Biogeochemistry NON ASSIMILATIVE Hindcast (PISCES) URL: http://marine.copernicus.eu/ |
| temp | Sea water temperature | degrees Celcius | Global Ocean Physics Reanalysis ECMWF ORAP5.0 (1979–2013) URL: http://marine.copernicus.eu/ |
| phybio | Carbon phytoplankton biomass | μmol/m$^2$ | Global Ocean Biogeochemistry NON ASSIMILATIVE Hindcast (PISCES) URL: http://marine.copernicus.eu/ |
| prod | Primary production | g/m$^2$/day | Global Ocean Biogeochemistry NON ASSIMILATIVE Hindcast (PISCES) URL: http://marine.copernicus.eu/ |
| Salinity | Sea water salinity | PSS | Global Ocean Physics Reanalysis ECMWF ORAP5.0 (1979–2013) URL: http://marine.copernicus.eu/ |
| SiO$_4$ | Silicate concentration | μmol/m$^2$ | Global Ocean Biogeochemistry NON ASSIMILATIVE Hindcast (PISCES) URL: http://marine.copernicus.eu/ |

**Note:**
Environmental parameters initially extracted from the BIO-ORACLE 2.0 database. All parameters are long-term maxima at minimum depth, except bathymetry, which represents the deepest (=minimum) depth measured.

We kept salinity although it was correlated with temperature as it is an important parameter to structuring deep-sea communities around Iceland (*e.g.*, *Weisshappel & Svavarsson, 1998*).

## Environmental cluster analysis

We hypothesized that deep-sea regions with similar environmental conditions would have a similar benthic fauna. Hence, we clustered the hexagonal grid cells based on the reduced set of the averaged environmental layers into a small set of environmentally homogenous regions. We used the *mclust* package (*Scrucca et al., 2016*) to conduct model-based hierarchical clustering using finite Gaussian Mixtures. The clustering algorithm compares 14 differently shaped types of Gaussian covariance structures representing different kinds of elliptical shapes ordered by an increasing complexity. The different models are compared using the Bayesian Information Criterion (*Burnham & Anderson, 2002*) choosing the model with the lowest complexity. Based on the plot of the different BIC models for possible cluster sizes from 2 to 10 (S3), we identified the optimal cluster as that one with highest regionalization capacity, *i.e.*, having a low number of clusters but already touching the plateau of the curve, signalling little differences in the model fit. We further confirmed the optimal number of clusters using a bootstrapped sequential likelihood ratio

test (*Scrucca et al., 2016*) by comparing an increasing number of cluster sizes. Finally, we calculated mean, standard deviation, minimum and maxima for each parameter and cluster combination. This was done to allow an interpretation of the environmental conditions representing the clusters.

## Taxonomic data

To interpret the overlap between clusters in terms of species composition, we first performed a constrained analysis of principal coordinates (CAP) (*Anderson & Willis, 2003*) with presence absence information and the Jaccard distance measure. CAP is an ordination technique, that allows to visualize similarities in sites based on species composition and environmental correlates. The ordination diagram was visually inspected by plotting the sites encircled by hulls on the first two axes. We further calculated the ANOSIM statistic on presence/absence transformed species data. ANOSIM is a non-parametric method to measure the community-wise overlap between different clusters (*Clarke, 1993*). It yields a statistic called R that is in the range from 0 to 1 with values of R below 0.5 indicating strong overlap. The statistic is tested for significance using a permuted $p$-value ($n = 9,999$). R-values above 0.75 indicate largely non-overlapping clusters with strongly different species composition. Both analyses were performed using the vegan package (*Oksanen et al., 2019*).

To identify characteristic species for each cluster, we identified all species being positively associated with one specific cluster or combinations of clusters using the *multipatt* function of the *indicspecies* package (*Cáceres & Legendre, 2009*). We used the group-size corrected Indicator Value (IndVal.g) as a measure of association. The null hypothesis tested is that the association of a species is not higher in a specific cluster than in the other clusters. This function calculates a $p$-value based on 9,999 permutations, which is not corrected for multiple testing. However, as we are not interested in the number of indicator species, but in whether a species has a high association to a cluster or not, the $p$-values do not have to be adjusted (*De Cáceres, Legendre & Moretti, 2010*). After the analysis, species with high association values were extracted as lists for each cluster combination. The resulting species-cluster relationship was compared with literature and information from the World Register of Marine Species (WoRMS) database (*Horton et al., 2021*).

## Diversity

We aimed to compare amphipod diversity between the different clusters. However, due to different numbers of samples ($n = 136$), *i.e.*, hexagonal cells, that contained the species data, clusters were not directly comparable in terms of diversity. Hence, we conducted a combined rarefaction-extrapolation analysis based on Hill numbers (*Chao, Chiu & Jost, 2014*). The concept of comparing species diversity using Hill numbers stems from the fact that most diversity indices are measures of entropy, such as Shannon or Simpson and do not translate directly into a diversity measure although often applied in such a way (*Jost, 2006*). Yet three well known measures of diversity *i.e.*, species richness, Shannon and Simpson diversity (*Shannon & Weaver, 1949*; *Simpson, 1949*) can be generalized by a

formula derived by Hill (*Jost, 2006*; *Chao, Chiu & Jost, 2014*) which orders the indices along an order of q, *i.e.*, q = 0, 1, 2 translating to richness, Shannon and Simpson, respectively. This order reflects an increasing importance of the evenness component of diversity, while the richness component becomes less effective. This means that for richness, there is no effect of abundance on the diversity measure, while for the Simpson index, rare species only have little effect on the estimated diversity values. Hence, the Simpson index is often thought to be the most robust index, when number of individuals strongly differ, as is the case here. The diversity information is transformed into a common measure of diversity, the effective number of species, which is the number of species having equal abundances that would be required to reach *e.g.*, the Shannon entropy value of the sample. This measure allows comparisons of all three different indices having the same unit, the effective number of species. We performed the analysis using the iNEXT package (*Hsieh, Ma & Chao, 2016*) based on the summed abundance vectors per species and cluster.

When studying deep-sea organisms, the most important indirect environmental gradient is depth in meters. In order to evaluate the diversity pattern related to depth we studied the original data from the different stations ($n = 532$) and expeditions. First, we calculated a Poisson Generalized Linear Model (GLM) to quantify the relationship between the number of species per station and depth in meters. Then we split the depth gradient in 100-m intervals to study the trend of the maximum number of species across the depth gradient.

## RESULTS

### General

The total number of amphipod individuals analyzed is $n = 71,108$. The assembled dataset contained 355 species from 141 genera and 44 families (Tab. 2). From these, 101 species were only be identified to the genus level, where species were given a numerical code. The original number of stations from the expeditions ($n = 532$) were reduced to a set of 136 one-degree wide hexagonal cells in order to reduce the heterogeneity in the dataset. These hexagonal cells were clustered according to their environmental conditions. The entire dataset is available *via* Peer J supplement as well as Pangaea (GfBIO) https://doi.pangaea.de/10.1594/PANGAEA.931959 (*Lörz et al., 2021*).

### Environmental clusters

The *mclust* algorithm identified six clusters to be the optimal configuration according to BIC and the likelihood ratio tests. However, when aggregating the species data to six clusters, this would result in clusters with disproportionally large differences in samples per cluster. Hence, we reduced the final number of clusters to four (Fig. 1). As the clustering is hierarchical, and the four-cluster solution is not much worse in terms of BIC we were confident that this aggregation is more informative with regard to the species than the six-cluster solution which would have split the northern and southern clusters into separate regions for the specific basins (the six cluster map is shown in the Supplementary Data). The four-cluster solution also provides a good overview of the large-scale spatial
**Table 2 Amphipoda species.**

| Nr | Species | Authority | Family |
|----|---------|-----------|--------|
| 1 | *Abludomelita gladiosa* | (Spence Bate, 1862) | Melitidae |
| 2 | *Abludomelita obtusata* | (Montagu, 1813) | Melitidae |
| 3 | *Acanthonotozoma cristatum* | (Ross, 1835) | Acanthonotozomatidae |
| 4 | *Acanthonotozoma serratum* | (O. Fabricius, 1780) | Acanthonotozomatidae |
| 5 | *Acanthostepheia malmgreni* | (Goës, 1866) | Oedicerotidae |
| 6 | *Aceroides latipes* | (G.O. Sars, 1883) | Oedicerotidae |
| 7 | *Aeginella spinosa* | Boeck, 1861 | Caprellidae |
| 8 | *Aeginina longicornis* | (Krøyer, 1843) | Caprellidae |
| 9 | *Ambasia atlantica* | (H. Milne Edwards, 1830) | Ambasiidae |
| 10 | *Ampelisca aequicornis* | Bruzelius, 1859 | Ampeliscidae |
| 11 | *Ampelisca amblyops* | G.O. Sars, 1891 | Ampeliscidae |
| 12 | *Ampelisca compacta* | Norman, 1882 | Ampeliscidae |
| 13 | *Ampelisca eschrichtii* | Krøyer, 1842 | Ampeliscidae |
| 14 | *Ampelisca gibba* | G.O. Sars, 1883 | Ampeliscidae |
| 15 | *Ampelisca islandica* | Bellan-Santini & Dauvin, 1996 | Ampeliscidae |
| 16 | *Ampelisca macrocephala* | Liljeborg, 1852 | Ampeliscidae |
| 17 | *Ampelisca odontoplax* | G. O. Sars, 1879 | Ampeliscidae |
| 18 | *Ampelisca* sp. A | Krøyer, 1842 | Ampeliscidae |
| 19 | *Ampelisca* sp. B | Krøyer, 1842 | Ampeliscidae |
| 20 | *Ampelisca uncinata* | Chevreux, 1887 | Ampeliscidae |
| 21 | *Amphilochoides boecki* | G.O. Sars, 1892 | Amphilochidae |
| 22 | *Amphilochoides serratipes* | (Norman, 1869) | Amphilochidae |
| 23 | *Amphilochus anoculus* | Tandberg & Vader, 2018 | Amphilochidae |
| 24 | *Amphilochus hamatus* | (Stephensen, 1925) | Amphilochidae |
| 25 | *Amphilochus manudens* | Spence Bate, 1862 | Amphilochidae |
| 26 | *Amphilochus* sp. A | Spence Bate, 1862 | Amphilochidae |
| 27 | *Amphilochus* sp. B | Spence Bate, 1862 | Amphilochidae |
| 28 | *Amphilochus* sp. C | Spence Bate, 1862 | Amphilochidae |
| 29 | *Amphilochus tenuimanus* | Boeck, 1871 | Amphilochidae |
| 30 | *Amphithopsis longicaudata* | Boeck, 1861 | Calliopiidae |
| 31 | *Andaniella pectinata* | G.O. Sars, 1883 | Stegocephalidae |
| 32 | *Andaniexis abyssi* | (Boeck, 1871) | Stegocephalidae |
| 33 | *Andaniexis lupus* | Berge & Vader, 1997 | Stegocephalidae |
| 34 | *Andaniexis* sp. A | Stebbing, 1906 | Stegocephalidae |
| 35 | *Andaniopsis nordlandica* | (Boeck, 1871) | Stegocephalidae |
| 36 | *Andaniopsis pectinata* | (G.O. Sars, 1883) | Stegocephalidae |
| 37 | *Anonyx* sp. A | Krøyer, 1838 | Uristidae |
| 38 | *Apherusa glacialis* | (Hansen, 1888) | Calliopiidae |
| 39 | *Apherusa sarsii* | Shoemaker, 1930 | Calliopiidae |
| 40 | *Apherusa* sp. A | Walker, 1891 | Calliopiidae |
| 41 | *Apherusa* sp. B | Walker, 1891 | Calliopiidae |

| Nr | Species | Authority | Family |
|----|---------|-----------|--------|
| 42 | *Apherusa* sp. C | Walker, 1891 | Calliopiidae |
| 43 | *Apherusa* sp. D | Walker, 1891 | Calliopiidae |
| 44 | *Argissa hamatipes* | (Norman, 1869) | Argissidae |
| 45 | *Arrhinopsis* sp. A | Stappers, 1911 | Oedicerotidae |
| 46 | *Arrhis phyllonyx* | (Sars, 1858) | Oedicerotidae |
| 47 | *Arrhis* sp. A | Stebbing, 1906 | Oedicerotidae |
| 48 | *Astyra abyssi* | Boeck, 1871 | Stilipedidae |
| 49 | *Astyra* sp. A | Boeck, 1871 | Stilipedidae |
| 50 | *Austrosyrrhoe septentrionalis* | Stephensen, 1931 | Synopiidae |
| 51 | *Austrosyrrhoe* sp. A | K.H. Barnard, 1925 | Synopiidae |
| 52 | *Autonoe borealis* | (Myers, 1976) | Aoridae |
| 53 | *Bathymedon longimanus* | (Boeck, 1871) | Oedicerotidae |
| 54 | *Bathymedon obtusifrons* | (Hansen, 1883) | Oedicerotidae |
| 55 | *Bathymedon saussurei* | (Boeck, 1871) | Oedicerotidae |
| 56 | *Bathymedon* sp. A | G.O. Sars, 1892 | Oedicerotidae |
| 57 | *Bruzelia* sp. A | Boeck, 1871 | Synopiidae |
| 58 | *Bruzelia tuberculata* | G.O. Sars, 1883 | Synopiidae |
| 59 | *Byblis crassicornis* | Metzger, 1875 | Ampeliscidae |
| 60 | *Byblis erythrops* | G.O. Sars, 1883 | Ampeliscidae |
| 61 | *Byblis gaimardii* | (Krøyer, 1846) | Ampeliscidae |
| 62 | *Byblis medialis* | Mills, 1971 | Ampeliscidae |
| 63 | *Byblis minuticornis* | Sars, 1879 | Ampeliscidae |
| 64 | *Byblis* sp. A | Boeck, 1871 | Ampeliscidae |
| 65 | *Byblisoides bellansantiniae* | Peart, 2018 | Ampeliscidae |
| 66 | *Calliopius laeviusculus* | (Krøyer, 1838) | Calliopiidae |
| 67 | *Camacho faroensis* | Myers, 1998 | Aoridae |
| 68 | *Caprella ciliata* | G.O. Sars, 1883 | Caprellidae |
| 69 | *Caprella dubia* | Hansen, 1887 | Caprellidae |
| 70 | *Caprella microtuberculata* | G. O. Sars, 1879 | Caprellidae |
| 71 | *Caprella rinki* | Stephensen, 1916 | Caprellidae |
| 72 | *Caprella septentrionalis* | Krøyer, 1838 | Caprellidae |
| 73 | *Chevreuxius grandimanus* | Bonnier, 1896 | Aoridae |
| 74 | *Cleippides bicuspis* | Stephensen, 1931 | Calliopiidae |
| 75 | *Cleippides quadricuspis* | Heller, 1875 | Calliopiidae |
| 76 | *Cleippides tricuspis* | (Krøyer, 1846) | Calliopiidae |
| 77 | *Cleonardo* sp. A | Stebbing, 1888 | Eusiridae |
| 78 | *Cleonardopsis* sp. A | K.H. Barnard, 1916 | Amathillopsidae |
| 79 | *Corophiidira* sp. A | Leach, 1814 (*sensu* Lowry & Myers, 2013) | Corophiidira |
| 80 | *Cressa carinata* | Stephensen, 1931 | Cressidae |
| 81 | *Cressa jeanjusti* | Krapp-Schickel, 2005 | Cressidae |
| 82 | *Cressa minuta* | Boeck, 1871 | Cressidae |

*(Continued)*

| Nr | Species | Authority | Family |
|---|---|---|---|
| 83 | *Cressa quinquedentata* | Stephensen, 1931 | Cressidae |
| 84 | *Cressina monocuspis* | Stephensen, 1931 | Cressidae |
| 85 | *Deflexilodes norvegicus* | (Boeck, 1871) | Oedicerotidae |
| 86 | *Deflexilodes rostratus* | (Stephensen, 1931) | Oedicerotidae |
| 87 | *Deflexilodes subnudus* | (Norman, 1889) | Oedicerotidae |
| 88 | *Deflexilodes tenuirostratus* | (Boeck, 1871) | Oedicerotidae |
| 89 | *Deflexilodes tesselatus* | (Schneider, 1883) | Oedicerotidae |
| 90 | *Deflexilodes tuberculatus* | (Boeck, 1871) | Oedicerotidae |
| 91 | *Dulichia* sp. A | Krøyer, 1845 | Dulichiidae |
| 92 | *Dulichia spinosissima* | Krøyer, 1845 | Dulichiidae |
| 93 | *Dulichiopsis* sp. A | Laubitz, 1977 | Dulichiidae |
| 94 | *Dyopedos porrectus* | Spence Bate, 1857 | Dulichiidae |
| 95 | *Dyopedos* sp. A | Spence Bate, 1857 | Dulichiidae |
| 96 | *Epimeria (Epimeria) loricata* | G.O. Sars, 1879 | Epimeriidae |
| 97 | *Epimeria (Epimeria)* sp. A | Costa in Hope, 1851 | Epimeriidae |
| 98 | *Ericthonius megalops* | (Sars G.O., 1879) | Ischyroceridae |
| 99 | *Eusirella elegans* | Chevreux, 1908 | Eusiridae |
| 100 | *Eusirogenes* sp. A | Stebbing, 1904 | Eusiridae |
| 101 | *Eusirogenes* sp. B | Stebbing, 1904 | Eusiridae |
| 102 | *Eusirus bathybius* | Schellenberg, 1955 | Eusiridae |
| 103 | *Eusirus biscayensis* | Bonnier, 1896 | Eusiridae |
| 104 | *Eusirus holmii* | Hansen, 1887 | Eusiridae |
| 105 | *Eusirus longipes* | Boeck, 1861 | Eusiridae |
| 106 | *Eusirus minutus* | G.O. Sars, 1893 | Eusiridae |
| 107 | *Eusirus propinquus* | Sars, 1893 | Eusiridae |
| 108 | *Eusirus* sp. A | Krøyer, 1845 | Eusiridae |
| 109 | *Eusirus* sp. B | Krøyer, 1845 | Eusiridae |
| 110 | *Eusirus* sp. C | Krøyer, 1845 | Eusiridae |
| 111 | *Eusirus* sp. D | Krøyer, 1845 | Eusiridae |
| 112 | *Eusyrophoxus* sp. A | Gurjanova, 1977 | Phoxocephalidae |
| 113 | *Gammaropsis* sp. A | Lilljeborg, 1855 | Photidae |
| 114 | *Gitana abyssicola* | G.O. Sars, 1892 | Amphilochidae |
| 115 | *Gitana rostrata* | Boeck, 1871 | Amphilochidae |
| 116 | *Gitana sarsi* | Boeck, 1871 | Amphilochidae |
| 117 | *Gitana* sp. A | Boeck, 1871 | Amphilochidae |
| 118 | *Gitanopsis arctica* | G.O. Sars, 1892 | Amphilochidae |
| 119 | *Gitanopsis bispinosa* | (Boeck, 1871) | Amphilochidae |
| 120 | *Gitanopsis inermis* | (G.O. Sars, 1883) | Amphilochidae |
| 121 | *Gitanopsis* sp. A | G.O. Sars, 1892 | Amphilochidae |
| 122 | *Glorandaniotes eilae* | (Berge & Vader, 1997) | Stegocephalidae |
| 123 | *Gronella groenlandica* | (Hansen, 1888) | Tryphosidae |

| Nr | Species | Authority | Family |
|---|---|---|---|
| 124 | *Halice abyssi* | Boeck, 1871 | Pardaliscidae |
| 125 | *Halice* sp. A | Boeck, 1871 | Pardaliscidae |
| 126 | *Halicoides* sp. A | Walker, 1896 | Pardaliscidae |
| 127 | *Halicoides tertia* | (Stephensen, 1931) | Pardaliscidae |
| 128 | *Halirages fulvocinctus* | (M. Sars, 1858) | Calliopiidae |
| 129 | *Halirages qvadridentatus* | G.O. Sars, 1877 | Calliopiidae |
| 130 | *Halirages* sp. A | Boeck, 1871 | Calliopiidae |
| 131 | *Haliragoides inermis* | (G.O. Sars, 1883) | Calliopiidae |
| 132 | *Haploops carinata* | Liljeborg, 1856 | Ampeliscidae |
| 133 | *Haploops dauvini* | Peart, 2018 | Ampeliscidae |
| 134 | *Haploops islandica* | Kaïm-Malka, Bellan-Santini & Dauvin, 2016 | Ampeliscidae |
| 135 | *Haploops kaimmalkai* | Peart, 2018 | Ampeliscidae |
| 136 | *Haploops setosa* | Boeck, 1871 | Ampeliscidae |
| 137 | *Haploops similis* | Stephensen, 1925 | Ampeliscidae |
| 138 | *Haploops* sp. A | Liljeborg, 1856 | Ampeliscidae |
| 139 | *Haploops tenuis* | Kanneworff, 1966 | Ampeliscidae |
| 140 | *Haploops tubicola* | Liljeborg, 1856 | Ampeliscidae |
| 141 | *Hardametopa nasuta* | (Boeck, 1871) | Stenothoidae |
| 142 | *Harpinia abyssi* | G.O. Sars, 1879 | Phoxocephalidae |
| 143 | *Harpinia antennaria* | Meinert, 1890 | Phoxocephalidae |
| 144 | *Harpinia crenulata* | (Boeck, 1871) | Phoxocephalidae |
| 145 | *Harpinia crenuloides* | Stephensen, 1925 | Phoxocephalidae |
| 146 | *Harpinia laevis* | Sars, 1891 | Phoxocephalidae |
| 147 | *Harpinia mucronata* | G. O. Sars, 1879 | Phoxocephalidae |
| 148 | *Harpinia pectinata* | Sars, 1891 | Phoxocephalidae |
| 149 | *Harpinia propinqua* | Sars, 1891 | Phoxocephalidae |
| 150 | *Harpinia* sp. A | Boeck, 1876 | Phoxocephalidae |
| 151 | *Harpinia* sp. B | Boeck, 1876 | Phoxocephalidae |
| 152 | *Harpinia* sp. C | Boeck, 1876 | Phoxocephalidae |
| 153 | *Harpinia* sp. D | Boeck, 1876 | Phoxocephalidae |
| 154 | *Harpinia* sp. E | Boeck, 1876 | Phoxocephalidae |
| 155 | *Harpinia* sp. F | Boeck, 1876 | Phoxocephalidae |
| 156 | *Harpinia* sp. G | Boeck, 1876 | Phoxocephalidae |
| 157 | *Harpinia* sp. H | Boeck, 1876 | Phoxocephalidae |
| 158 | *Harpinia truncata* | Sars, 1891 | Phoxocephalidae |
| 159 | *Harpiniopsis similis* | Stephensen, 1925 | Phoxocephalidae |
| 160 | *Hippomedon gorbunovi* | Gurjanova, 1929 | Tryphosidae |
| 161 | *Hippomedon propinqvus* | G.O. Sars, 1890 | Tryphosidae |
| 162 | *Idunella aeqvicornis* | (G.O. Sars, 1877) | Liljeborgiidae |
| 163 | *Idunella* sp. A | G.O. Sars, 1894 | Liljeborgiidae |
| 164 | *Ischyrocerus anguipes* | Krøyer, 1838 | Ischyroceridae |

(Continued)

| Nr | Species | Authority | Family |
|---|---|---|---|
| 165 | *Ischyrocerus latipes* | Krøyer, 1842 | Ischyroceridae |
| 166 | *Ischyrocerus megacheir* | (Boeck, 1871) | Ischyroceridae |
| 167 | *Ischyrocerus megalops* | Sars, 1894 | Ischyroceridae |
| 168 | *Jassa* sp. A | Leach, 1814 | Ischyroceridae |
| 169 | *Kerguelenia borealis* | G.O. Sars, 1891 | Kergueleniidae |
| 170 | *Laetmatophilus* sp. A | Bruzelius, 1859 | Podoceridae |
| 171 | *Laetmatophilus tuberculatus* | Bruzelius, 1859 | Podoceridae |
| 172 | *Laothoes meinerti* | Boeck, 1871 | Calliopiidae |
| 173 | *Laothoes pallaschi* | Coleman, 1999 | Calliopiidae |
| 174 | *Laothoes* sp. A | Boeck, 1871 | Calliopiidae |
| 175 | *Lepechinella arctica* | Schellenberg, 1926 | Lepechinellidae |
| 176 | *Lepechinella grimi* | Thurston, 1980 | Lepechinellidae |
| 177 | *Lepechinella helgii* | Thurston, 1980 | Lepechinellidae |
| 178 | *Lepechinella skarphedini* | Thurston, 1980 | Lepechinellidae |
| 179 | *Lepechinella victoriae* | Johansen & Vader, 2015 | Lepechinellidae |
| 180 | *Lepechinelloides karii* | Thurston, 1980 | Lepechinellidae |
| 181 | *Lepidepecreum* sp. A | Spence Bate & Westwood, 1868 | Tryphosidae |
| 182 | *Leptamphopus sarsi* | Vanhöffen, 1897 | Calliopiidae |
| 183 | *Leptophoxus falcatus* | (G.O. Sars, 1883) | Phoxocephalidae |
| 184 | *Leucothoe lilljeborgi* | Boeck, 1861 | Leucothoidae |
| 185 | *Leucothoe* sp. A | Leach, 1814 | Leucothoidae |
| 186 | *Leucothoe spinicarpa* | (Abildgaard, 1789) | Leucothoidae |
| 187 | *Leucothoe vaderotti* | Krapp-Schickel, 2018 | Leucothoidae |
| 188 | *Liljeborgia fissicornis* | (Sars, 1858) | Liljeborgiidae |
| 189 | *Liljeborgia pallida* | (Spence Bate, 1857) | Liljeborgiidae |
| 190 | *Liljeborgia* sp. A | Spence Bate, 1862 | Liljeborgiidae |
| 191 | *Lysianella petalocera* | G.O. Sars, 1883 | Tryphosidae |
| 192 | *Megamoera dentata* | (Krøyer, 1842) | Melitidae |
| 193 | *Megamphopus raptor* | Myers, 1998 | Photidae |
| 194 | *Melphidippa borealis* | Boeck, 1871 | Melphidippidae |
| 195 | *Melphidippa goesi* | Stebbing, 1899 | Melphidippidae |
| 196 | *Melphidippa macrura* | G.O. Sars, 1894 | Melphidippidae |
| 197 | *Melphidippa* sp. A | Boeck, 1871 | Melphidippidae |
| 198 | *Melphidippa* sp. B | Boeck, 1871 | Melphidippidae |
| 199 | *Metacaprella horrida* | (Sars G.O., 1877) | Caprellidae |
| 200 | *Metandania wimi* | Berge, 2001 | Stegocephalidae |
| 201 | *Metopa abyssalis* | Stephensen, 1931 | Stenothoidae |
| 202 | *Metopa boeckii* | Sars, 1892 | Stenothoidae |
| 203 | *Metopa bruzelii* | (Goës, 1866) | Stenothoidae |
| 204 | *Metopa leptocarpa* | G.O. Sars, 1883 | Stenothoidae |
| 205 | *Metopa norvegica* | (Liljeborg, 1851) | Stenothoidae |

| Nr | Species | Authority | Family |
|---|---|---|---|
| 206 | *Metopa palmata* | Sars, 1892 | Stenothoidae |
| 207 | *Metopa robusta* | Sars, 1892 | Stenothoidae |
| 208 | *Metopa rubrovittata* | G.O. Sars, 1883 | Stenothoidae |
| 209 | *Metopa sinuata* | Sars, 1892 | Stenothoidae |
| 210 | *Metopa* sp. A | Boeck, 1871 | Stenothoidae |
| 211 | *Metopa* sp. B | Boeck, 1871 | Stenothoidae |
| 212 | *Metopa* sp. C | Boeck, 1871 | Stenothoidae |
| 213 | *Metopa* sp. D | Boeck, 1871 | Stenothoidae |
| 214 | *Metopa* sp. E | Boeck, 1871 | Stenothoidae |
| 215 | *Monoculodes latimanus* | (Goës, 1866) | Oedicerotidae |
| 216 | *Monoculodes latissimanus* | Stephensen, 1931 | Oedicerotidae |
| 217 | *Monoculodes packardi* | Boeck, 1871 | Oedicerotidae |
| 218 | *Monoculodes* sp. A | Stimpson, 1853 | Oedicerotidae |
| 219 | *Monoculodes* sp. B | Stimpson, 1853 | Oedicerotidae |
| 220 | *Monoculodes* sp. C | Stimpson, 1853 | Oedicerotidae |
| 221 | *Monoculodes* sp. D | Stimpson, 1853 | Oedicerotidae |
| 222 | *Monoculodes* sp. E | Stimpson, 1853 | Oedicerotidae |
| 223 | *Monoculodes* sp. F | Stimpson, 1853 | Oedicerotidae |
| 224 | *Monoculodes* sp. G | Stimpson, 1853 | Oedicerotidae |
| 225 | *Monoculopsis longicornis* | (Boeck, 1871) | Oedicerotidae |
| 226 | *Neopleustes boecki* | (Hansen, 1888) | Pleustidae |
| 227 | *Neopleustes pulchellus* | (Krøyer, 1846) | Pleustidae |
| 228 | *Neopleustes* sp. A | Stebbing, 1906 | Pleustidae |
| 229 | *Nicippe tumida* | Bruzelius, 1859 | Pardaliscidae |
| 230 | *Nototropis smitti* | (Goës, 1866) | Atylidae |
| 231 | *Nototropis* sp. A | Costa, 1853 | Atylidae |
| 232 | *Odius carinatus* | (Spence Bate, 1862) | Ochlesidae |
| 233 | *Oedicerina ingolfi* | Stephensen, 1931 | Oedicerotidae |
| 234 | *Oedicerina* sp. A | Stephensen, 1931 | Oedicerotidae |
| 235 | *Oediceropsis brevicornis* | Lilljeborg, 1865 | Oedicerotidae |
| 236 | *Oediceropsis* sp. A | Lilljeborg, 1865 | Oedicerotidae |
| 237 | *Oediceros borealis* | Boeck, 1871 | Oedicerotidae |
| 238 | *Onisimus plautus* | (Krøyer, 1845) | Uristidae |
| 239 | *Orchomene macroserratus* | Shoemaker, 1930 | Tryphosidae |
| 240 | *Orchomene pectinatus* | G.O. Sars, 1883 | Tryphosidae |
| 241 | *Orchomene* sp. A | Boeck, 1871 | Tryphosidae |
| 242 | *Pacificulodes pallidus* | (G.O. Sars, 1892) | Oedicerotidae |
| 243 | *Paradulichia typica* | Boeck, 1871 | Dulichiidae |
| 244 | *Paramphilochoides odontonyx* | (Boeck, 1871) | Amphilochidae |
| 245 | *Paramphithoe hystrix* | (Ross, 1835) | Paramphithoidae |
| 246 | *Parandania gigantea* | (Stebbing, 1883) | Stegocephalidae |

(Continued)

| Nr | Species | Authority | Family |
|---|---|---|---|
| 247 | *Paraphoxus oculatus* | (G. O. Sars, 1879) | Phoxocephalidae |
| 248 | *Parapleustes assimilis* | (G.O. Sars, 1883) | Pleustidae |
| 249 | *Parapleustes bicuspis* | (Krøyer, 1838) | Pleustidae |
| 250 | *Pardalisca abyssi* | Boeck, 1871 | Pardaliscidae |
| 251 | *Pardalisca cuspidata* | Krøyer, 1842 | Pardaliscidae |
| 252 | *Pardalisca* sp. A | Krøyer, 1842 | Pardaliscidae |
| 253 | *Pardalisca* sp. B | Krøyer, 1842 | Pardaliscidae |
| 254 | *Pardalisca* sp. C | Krøyer, 1842 | Pardaliscidae |
| 255 | *Pardalisca tenuipes* | G.O. Sars, 1893 | Pardaliscidae |
| 256 | *Pardaliscoides tenellus* | Stebbing, 1888 | Pardaliscidae |
| 257 | *Paroediceros curvirostris* | (Hansen, 1888) | Oedicerotidae |
| 258 | *Paroediceros lynceus* | (M. Sars, 1858) | Oedicerotidae |
| 259 | *Paroediceros propinquus* | (Goës, 1866) | Oedicerotidae |
| 260 | *Perioculodes longimanus* | (Spence Bate & Westwood, 1868) | Oedicerotidae |
| 261 | *Phippsia gibbosa* | (G.O. Sars, 1883) | Stegocephalidae |
| 262 | *Phippsia roemeri* | Schellenberg, 1925 | Stegocephalidae |
| 263 | *Photis reinhardi* | Krøyer, 1842 | Photidae |
| 264 | *Phoxocephalus holbolli* | (Krøyer, 1842) | Phoxocephalidae |
| 265 | *Pleustes (Pleustes) panoplus* | (Krøyer, 1838) | Pleustidae |
| 266 | *Pleustes tuberculatus* | Spence Bate, 1858 | Pleustidae |
| 267 | *Pleusymtes pulchella* | (G.O. Sars, 1876) | Pleustidae |
| 268 | *Pleusymtes* sp. A | J.L. Barnard, 1969 | Pleustidae |
| 269 | *Pontocrates arcticus* | G.O. Sars, 1895 | Oedicerotidae |
| 270 | *Pontocrates* sp. A | Boeck, 1871 | Oedicerotidae |
| 271 | *Proaeginina norvegica* | (Stephensen, 1931) | Caprellidae |
| 272 | *Proboloides gregaria* | (G.O. Sars, 1883) | Stenothoidae |
| 273 | *Protellina ingolfi* | Stephensen, 1944 | Caprellidae |
| 274 | *Protoaeginella muriculata* | Laubitz & Mills, 1972 | Caprellidae |
| 275 | *Protomedeia fasciata* | Krøyer, 1842 | Corophiidae |
| 276 | *Pseudo bioice* | (Berge & Vader, 1997) | Stegocephalidae |
| 277 | *Pseudotiron* sp. A | Chevreux, 1895 | Synopiidae |
| 278 | *Rhachotropis aculeata* | (Lepechin, 1780) | Eusiridae |
| 279 | *Rhachotropis arii* | Thurston, 1980 | Eusiridae |
| 280 | *Rhachotropis distincta* | (Holmes, 1908) | Eusiridae |
| 281 | *Rhachotropis gislii* | Thurston, 1980 | Eusiridae |
| 282 | *Rhachotropis gloriosae* | Ledoyer, 1982 | Eusiridae |
| 283 | *Rhachotropis helleri* | (Boeck, 1871) | Eusiridae |
| 284 | *Rhachotropis inflata* | (G.O. Sars, 1883) | Eusiridae |
| 285 | *Rhachotropis kergueleni* | Stebbing, 1888 | Eusiridae |
| 286 | *Rhachotropis leucophthalma* | G.O. Sars, 1893 | Eusiridae |
| 287 | *Rhachotropis macropus* | G.O. Sars, 1893 | Eusiridae |

| Nr | Species | Authority | Family |
|---|---|---|---|
| 288 | *Rhachotropis northriana* | d'Udekem d'Acoz, Vader & Legezinska, 2007 | Eusiridae |
| 289 | *Rhachotropis oculata* | (Hansen, 1887) | Eusiridae |
| 290 | *Rhachotropis palporum* | Stebbing, 1908 | Eusiridae |
| 291 | *Rhachotropis proxima* | Chevreux, 1911 | Eusiridae |
| 292 | *Rhachotropis* sp. A | S.I. Smith, 1883 | Eusiridae |
| 293 | *Rhachotropis* sp. B | S.I. Smith, 1883 | Eusiridae |
| 294 | *Rhachotropis* sp. C | S.I. Smith, 1883 | Eusiridae |
| 295 | *Rhachotropis* sp. D | S.I. Smith, 1883 | Eusiridae |
| 296 | *Rhachotropis thordisae* | Thurston, 1980 | Eusiridae |
| 297 | *Rhachotropis thorkelli* | Thurston, 1980 | Eusiridae |
| 298 | *Rostroculodes borealis* | (Boeck, 1871) | Oedicerotidae |
| 299 | *Rostroculodes kroyeri* | (Boeck, 1870) | Oedicerotidae |
| 300 | *Rostroculodes longirostris* | (Goës, 1866) | Oedicerotidae |
| 301 | *Scopelocheirus* sp. A | Spence Bate, 1857 | Scopelocheiridae |
| 302 | *Sicafodia iceage* | Campean & Coleman, 2017 | Sicafodiidae |
| 303 | *Sicafodia* sp. A | Just, 2004 | Sicafodiidae |
| 304 | *Siphonoecetes typicus* | Krøyer, 1845 | Ischyroceridae |
| 305 | *Socarnes bidenticulatus* | (Spence Bate, 1858) | Lysianassidae |
| 306 | *Socarnes vahlii* | (Krøyer, 1838) | Lysianassidae |
| 307 | *Stegocephalina wagini* | (Gurjanova, 1936) | Stegocephalidae |
| 308 | *Stegocephaloides auratus* | (G.O. Sars, 1883) | Stegocephalidae |
| 309 | *Stegocephaloides barnardi* | Berge & Vader, 1997 | Stegocephalidae |
| 310 | *Stegocephaloides christianiensis* | Boeck, 1871 | Stegocephalidae |
| 311 | *Stegocephalus ampulla* | (Phipps, 1774) | Stegocephalidae |
| 312 | *Stegocephalus inflatus* | Krøyer, 1842 | Stegocephalidae |
| 313 | *Stegocephalus similis* | Sars, 1891 | Stegocephalidae |
| 314 | *Stegocephalus* sp. A | Krøyer, 1842 | Stegocephalidae |
| 315 | *Stegocephalus* sp. B | Krøyer, 1842 | Stegocephalidae |
| 316 | *Stegonomadia biofar* | (Berge & Vader, 1997) | Stegocephalidae |
| 317 | *Stegonomadia idae* | (Berge & Vader, 1997) | Stegocephalidae |
| 318 | *Stegoplax longirostris* | G.O. Sars, 1883 | Cyproideidae |
| 319 | *Stegoplax* sp. A | G.O. Sars, 1883 | Cyproideidae |
| 320 | *Stenopleustes latipes* | (M. Sars, 1858) | Pleustidae |
| 321 | *Stenopleustes malmgreni* | (Boeck, 1871) | Pleustidae |
| 322 | *Stenopleustes nodifera* | (G.O. Sars, 1883) | Pleustidae |
| 323 | *Stenopleustes* sp. A | G.O. Sars, 1893 | Pleustidae |
| 324 | *Stenothoe marina* | (Spence Bate, 1857) | Stenothoidae |
| 325 | *Stenothoe megacheir* | (Boeck, 1871) | Stenothoidae |
| 326 | *Stenothoe* sp. A | Dana, 1852 | Stenothoidae |
| 327 | *Stenothoe* sp. B | Dana, 1852 | Stenothoidae |
| 328 | *Stenothoe* sp. C | Dana, 1852 | Stenothoidae |

(Continued)

| Table 2 (continued) | | | |
|---|---|---|---|
| Nr | Species | Authority | Family |
| 329 | *Stenothoe* sp. D | Dana, 1852 | Stenothoidae |
| 330 | *Stephobruzelia dentata* | (Stephensen, 1931) | Synopiidae |
| 331 | *Synchelidium haplocheles* | (Grube, 1864) | Oedicerotidae |
| 332 | *Synchelidium intermedium* | Sars, 1892 | Oedicerotidae |
| 333 | *Synchelidium* sp. A | G.O. Sars, 1892 | Oedicerotidae |
| 334 | *Syrrhoe crenulata* | Goës, 1866 | Synopiidae |
| 335 | *Syrrhoe* sp. A | Goës, 1866 | Synopiidae |
| 336 | *Syrrhoites pusilla* | Enequist, 1949 | Synopiidae |
| 337 | *Syrrhoites serrata* | (G.O. Sars, 1879) | Synopiidae |
| 338 | *Syrrhoites* sp. A | G.O. Sars, 1893 | Synopiidae |
| 339 | *Themisto gaudichaudii* | Guérin, 1825 | Hyperiidae |
| 340 | *Thorina elongata* | Laubitz & Mills, 1972 | Caprellidae |
| 341 | *Thorina spinosa* | Stephensen, 1944 | Caprellidae |
| 342 | *Tiron spiniferus* | (Stimpson, 1853) | Synopiidae |
| 343 | *Tmetonyx cicada* | (Fabricius, 1780) | Uristidae |
| 344 | *Tmetonyx* sp. A | Stebbing, 1906 | Uristidae |
| 345 | *Tryphosella schneideri* | (Stephensen, 1921) | Tryphosidae |
| 346 | *Tryphosella* sp. A | Bonnier, 1893 | Tryphosidae |
| 347 | *Unciola laticornis* | Hansen, 1887 | Unciolidae |
| 348 | *Unciola leucopis* | (Krøyer, 1845) | Unciolidae |
| 349 | *Unciola planipes* | Norman, 1867 | Unciolidae |
| 350 | *Urothoe elegans* | Spence Bate, 1857 | Urothoidae |
| 351 | *Westwoodilla brevicalcar* | Goës, 1866 | Oedicerotidae |
| 352 | *Westwoodilla caecula* | (Spence Bate, 1857) | Oedicerotidae |
| 353 | *Westwoodilla megalops* | (G.O. Sars, 1883) | Oedicerotidae |
| 354 | *Westwoodilla* sp. A | Spence Bate, 1862 | Oedicerotidae |
| 355 | *Xenodice* sp. A | Boeck, 1871 | Podoceridae |

**Note:**
Amphipoda species, authorities and family.

pattern. There is a "Coastal" cluster ($n = 34$ cells) which is always close to the coastline and is characterized by shallow depth, high amounts of dissolved iron and phytobiomass and warm, oxygen-rich waters with a high current speed (Fig. 2). The second cluster resembles the GIFR ($n = 55$), which spreads from west to east and separates the northern and southern basis. In many points it is similar to the coastal cluster but is deeper and with less dissolved iron, oxygen, and phytobiomass. The other two clusters are called "Deep South" ($n = 19$) and "Deep North" ($n = 28$) as they represent the deep-sea regions around Iceland. They differ strongly from the first two clusters by having very low values for many parameters. "Deep North" differs from "Deep South" by being much colder, with almost no current velocity. Further, "Deep North" has a much higher amount of dissolved oxygen and pH. The lowest depths of around 3,400 m are observed in the Aegir ridge. These four

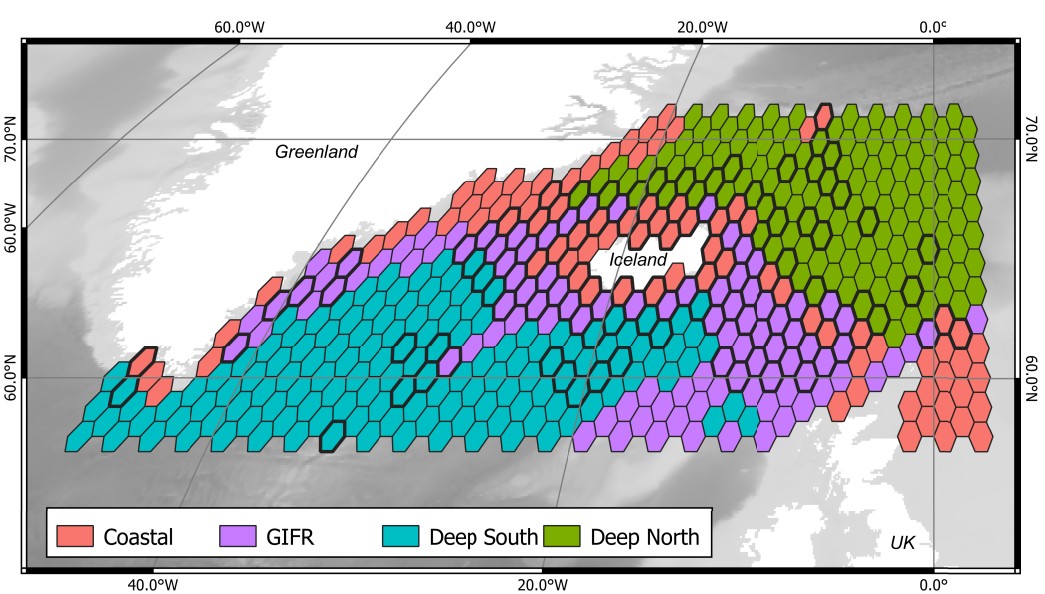

**Figure 1  Four environmental clusters.** Map of the outlines for four identified environmental clusters in the North Atlantic. The Greenland-Faroe-Iceland ridge (GIFR) extends from west to east and is, like the coastal cluster, partly interrupted due to the coarse resolution of the hexagonal cells of 1° in east-west direction, ($n$ = 469).        

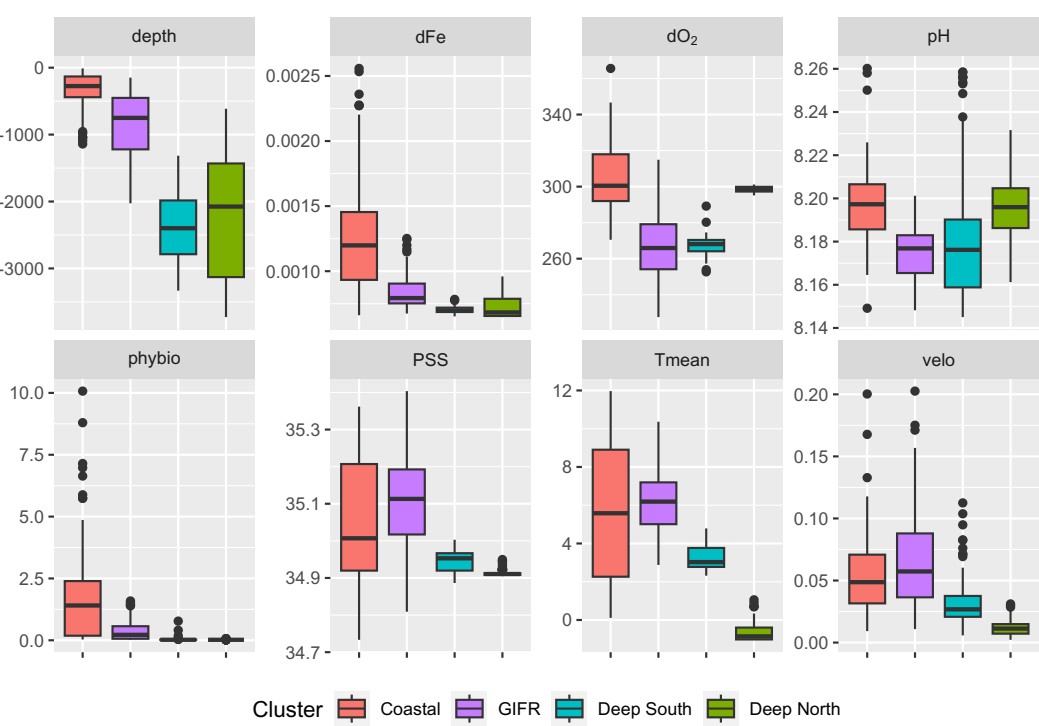

**Figure 2  Environmental parameters.** Characterization of the four environmental clusters by the environmental parameters with box-whisker plots. For abbreviations refer to Table 1. An extended table with numeric information can be found in the appendix.

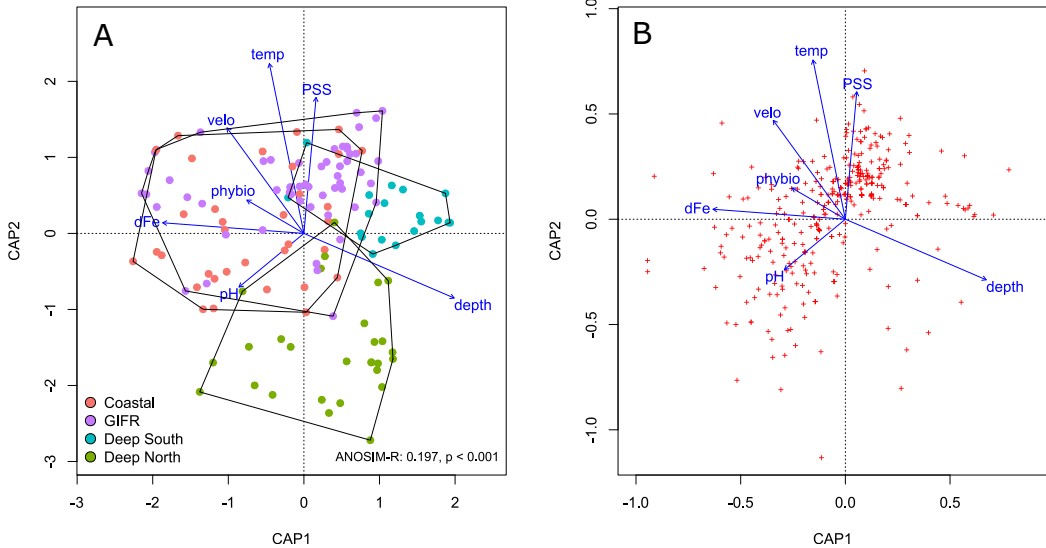

**Figure 3 Constrained analysis of principal coordinates.** Constrained analysis of principal coordinates (CAP) based on Jaccard distances. (A) Scaling is based on site scores, (B) scaling is based on species scores (red dots)—note the differences of the axes. Arrows point into the direction of largest correlation with species and site scores. The 0,0 coordinate reflects the centroid of each variable. The environmental clusters still overlap considerably in their species composition as reflected by the low ANOSIM-R statistic.

clusters thus characterize the environmental conditions around Iceland on a regional spatial scale.

## Constrained ordination

We conducted a constrained ordination to verify the amount of variation explained in the species data by the environmental information contained at the level of the hexagonal cells. The constrained axes of the ordination explained 11% of the total variation, while 89% is explained by the 357 unconstrained axes. According to a permutation test of the marginal effects of each environmental variable carried out using the *anova.cca* function of the *vegan* package, the most important environmental variables were temperature (F = 2.34, $p < 0.001$), depth (F = 2.123, $p < 0.001$), and salinity (F = 2.01, $p < 0.001$).

The four different clusters strongly overlapped in ordination space (Fig. 3A). The ANOSIM-R value of 0.197 signals considerable similarity in species composition between the clusters. All clusters overlap in the centre of the diagram; their large spread indicates strong heterogeneity. The deep-sea clusters overlapped less than the coastal and GIFR-cluster. In general, the first constrained axis represented the depth gradient, which was in contrast to all other variables. Salinity, temperature and pH characterized the second constrained axes, with pH being in contrast to temperature and salinity (Fig. 3A). The species pattern clumped near the centroid of the ordination diagram (Fig. 3B) indicating that many species are found in intermediate environmental conditions. Fewer species have a clear centroid in deeper waters, instead many species favour higher temperatures and an above average salinity. Large variation appears in the direction of pH and dissolved iron, as indicated by the strong scatter of species centroids (Fig. 3B).

## Indicator species analysis

To characterize the different clusters with regard to faithful species, *i.e.*, so-called indicator species we conducted a multipattern indicator species analysis. We compared 15 different combinations with an increasing number of clusters. From 355 species, we identified 56 to have a strong association to one or more clusters. Fourty-three species were associated to one cluster only, while twelve and one species were associated to two and three clusters, respectively (Table 3). Only two species were found for the GIFR cluster, but more species from GIFR appear in combination with other clusters.

Three of the clusters, the Deep North, the Deep South and the Coastal have indicator species belonging to the genus *Rhachotropis*. While different species of a genus might be specialized on different diets, all *Rhachotropis* species are very good swimmers (*Lörz, 2010*). The Deep South cluster has four *Rhachotropis* as indicator species. While the GIFR cluster only had two endobenthic species, belonging to the family Ampeliscidae, which are not considered strong swimmers (*Peart, 2018*), the combined GIFR and coastal cluster indicate *Rhachotropis aculeata* (*Lepechin, 1780*) as an indicator—a species that is known to have a circum-Arctic distribution (*Lörz et al., 2018*). *Caprella microtuberculata* G. O. *Sars, 1879* and *Aeginella spinosa Boeck, 1861* are indicator species of the combined coastal and GIFR cluster. These two species belong to the amphipod group Caprellidae, skeleton or ghost shrimps, which are known for their clinging lifestyle. The indicator species with the highest values, over 0.5, are *Cleippides quadricuspis Heller, 1875* from the Deep North, *Eusirus holmi Hansen, 1887* from the combined Coastal and Deep North cluster and *Rhachotropis thordisae Thurston, 1980* from the Deep South cluster—these three species are all large amphipods of several cm body length and known as predators (*Lörz et al., 2018*).

## Diversity

The number of aggregated hexagonal cells differed for each cluster, hence we had to apply a rarefaction and extrapolation analysis to make the three diversity measures comparable. The rarefaction of the summed abundances revealed that the two clusters "coastal" and "GIFR" have about twice the number of species than the deep-sea clusters (Fig. 4A). This even holds when only the lowest comparable value of approximate 10,000 individuals is considered. Although there were so many individuals per cluster, the curves do not level off, indicating that still more sampling would be required to reach a plateau in species richness. The Shannon diversity (Fig. 4B) considers the richness-abundance component of diversity. The "coastal" and "GIFR" clusters are at the same level of 60 effective species; the deep-sea clusters again have a much lower diversity, *i.e.*, almost three times lower. All curves reach a plateau, indicating that there is little more diversity to expect when abundances are considered. Hence, only rare species might be added by future sampling. Considering the Simpson diversity (Fig. 4C), *i.e.*, when no rare species but only dominant species have an influence on the diversity measure, then the "coastal" cluster becomes the most diverse cluster while the "GIFR" is only half as diverse as the coastal cluster.

The richness pattern across the depth gradient showed high variation at depths above 1,500 m with richness values up to 79 species per station (Fig. 5A). Most of the stations

**Table 3 Indicator value analysis for all combinations of the environmental clusters.** The group-size corrected Indicator Value (IndVal.g) represent the association value of a species with a given cluster. The $p$-value is based on 999 permutations. Asterisks code for $p$-values at signifcance levels of 5% (*) and 1% (**).

| Cluster | Nr. | Species | IndVal.g | $p$-value | |
|---|---|---|---|---|---|
| Coastal | 1 | *Rhachotropis oculata* | 0.400 | 0.005 | |
| | 2 | *Westwoodilla caecula* | 0.383 | 0.015 | |
| | 3 | *Ampelisca macrocephala* | 0.368 | 0.010 | ** |
| | 4 | *Deflexilodes tesselatus* | 0.368 | 0.035 | * |
| | 5 | *Harpinia* sp. E | 0.343 | 0.020 | * |
| | 6 | *Monoculodes* sp. A | 0.343 | 0.015 | * |
| | 7 | *Westwoodilla megalops* | 0.343 | 0.030 | * |
| | 8 | *Harpinia pectinata* | 0.328 | 0.020 | * |
| | 9 | *Bathymedon obtusifrons* | 0.319 | 0.035 | * |
| | 10 | *Monoculodes latimanus* | 0.297 | 0.045 | * |
| Deep North | 1 | *Cleippides quadricuspis* | 0.642 | 0.005 | ** |
| | 2 | *Bruzelia dentata* | 0.463 | 0.005 | ** |
| | 3 | *Rhachotropis* sp. A | 0.392 | 0.005 | ** |
| | 4 | *Paroediceros curvirostris* | 0.375 | 0.015 | * |
| | 5 | *Deflexilodes tenuirostratus* | 0.349 | 0.040 | * |
| | 6 | *Halirages quadridentata* | 0.344 | 0.025 | * |
| | 7 | *Monoculopsis longicornis* | 0.344 | 0.025 | * |
| | 8 | *Oedicerina* sp. | 0.327 | 0.025 | * |
| Deep South | 1 | *Rhachotropis thordisae* | 0.559 | 0.005 | ** |
| | 2 | *Rhachotropis proxima* | 0.499 | 0.010 | ** |
| | 3 | *Eusirus bathybius* | 0.459 | 0.010 | ** |
| | 4 | *Lepechinelloides karii* | 0.459 | 0.005 | ** |
| | 5 | *Rhachotropis gislii* | 0.459 | 0.005 | ** |
| | 6 | *Protoaeginella muriculata* | 0.401 | 0.010 | ** |
| | 7 | *Cleonardopsis* sp. | 0.397 | 0.005 | ** |
| | 8 | *Lepechinella grimi* | 0.397 | 0.005 | ** |
| | 9 | *Lepechinella helgii* | 0.397 | 0.010 | ** |
| | 10 | *Lepechinella skarphedini* | 0.397 | 0.010 | ** |
| | 11 | *Rhachotropis thorkelli* | 0.397 | 0.010 | ** |
| | 12 | *Neopleustes boecki* | 0.365 | 0.010 | ** |
| | 13 | *Neopleustes* sp. | 0.324 | 0.010 | ** |
| | 14 | *Sicafodia* sp. | 0.324 | 0.010 | ** |
| | 15 | *Eusirus* sp. C | 0.300 | 0.020 | * |
| | 16 | *Rhachotropis aislii* | 0.300 | 0.040 | * |
| | 17 | *Rhachotropis gloriosae* | 0.300 | 0.035 | * |
| GFIR | 1 | *Ampelisca odontoplax* | 0.348 | 0.03 | * |
| | 2 | *Haploops tenuis* | 0.302 | 0.05 | * |
| Coastal + Deep North | 1 | *Eusirus holmi* | 0.509 | 0.005 | ** |
| | 2 | *Halirages fulvocincta* | 0.490 | 0.050 | * |
| | 3 | *Arrhis phyllonyx* | 0.458 | 0.005 | ** |
| | 4 | *Andaniella pectinata* | 0.430 | 0.005 | ** |
| | 5 | *Paroediceros propinquus* | 0.372 | 0.040 | * |
| | 6 | *Halirages elegans* | 0.359 | 0.030 | * |
| | 7 | *Harpiniopsis similis* | 0.347 | 0.035 | * |

| Table 3 (continued) | | | | | |
|---|---|---|---|---|---|
| Cluster | Nr. | Species | IndVal.g | *p*-value | |
| Coastal + GFIR | 1 | *Aeginella spinosa* | 0.559 | 0.005 | ** |
| | 2 | *Rhachotropis aculeata* | 0.467 | 0.025 | * |
| | 3 | *Caprella microtuberculata* | 0.462 | 0.010 | ** |
| | 4 | *Harpinia propinqua* | 0.459 | 0.030 | * |
| Deep South + Deep North | 1 | *Liljeborgia pallida* | 0.349 | 0.045 | * |
| | 2 | *Ampelisca islandica* | 0.329 | 0.025 | * |
| Coastal + Deep South + Deep North | 1 | *Amphilochus anoculus* | 0.424 | 0.035 | * |

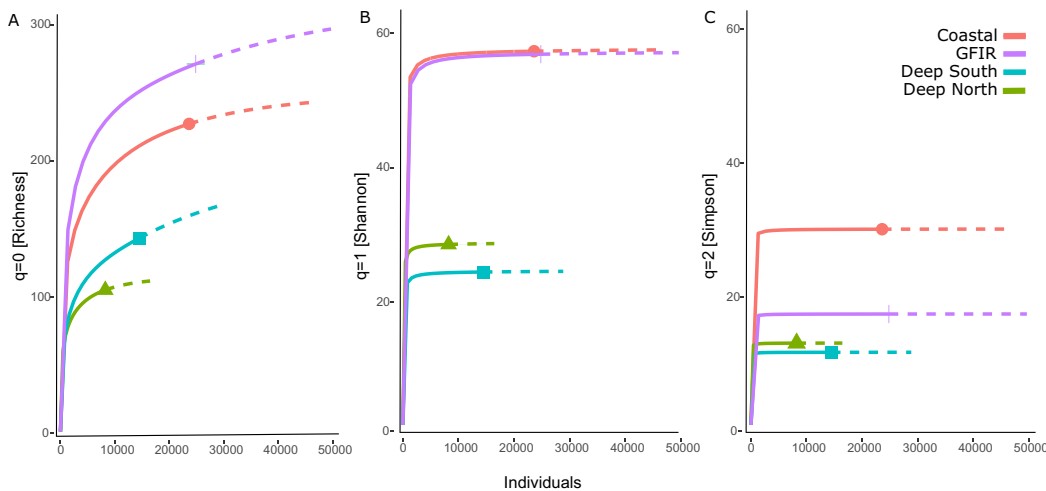

**Figure 4 Rarefaction-extrapolation of diversity indices per cluster.** The diversity indices (A) richness, (B) Shannon, and (C) Simpson, represent an increasing importance of abundant species. The unit of the y-axis is the effective number of species.

recorded rather few species *i.e.*, up to 10 species with an average of 20 species at the shallowest parts (18 m) and an estimated richness of eight species at the lowest depths. The trend for the maximum number of species aggregated per 100-m interval showed an unimodal pattern with a peak at depths around 500 m and a much lower richness at depths lower than 1,000 m (Fig. 5B). These figures support the finding that the Coastal and GIFR clusters are much more diverse than the deep-sea clusters (Fig. 5C).

# DISCUSSION

## Environmental and historical imprints on amphipod distributions

Distributional groupings given in the present study corresponded to earlier findings, in which distinctive boundaries between a northern and a southern deep-sea fauna were inferred, while the composition of the shallow-water fauna (<500 m) around Iceland was very similar (*Weisshappel & Svavarsson, 1998*; *Weisshappel, 2000*; *Bett, 2001*; *Weisshappel, 2001*). Unsurprisingly, the spatial distribution of amphipods appeared to be most strongly influenced by bathymetry, salinity and seafloor temperature. The latter two were interconnected and indicative of particular water masses (*Puerta et al., 2020*).

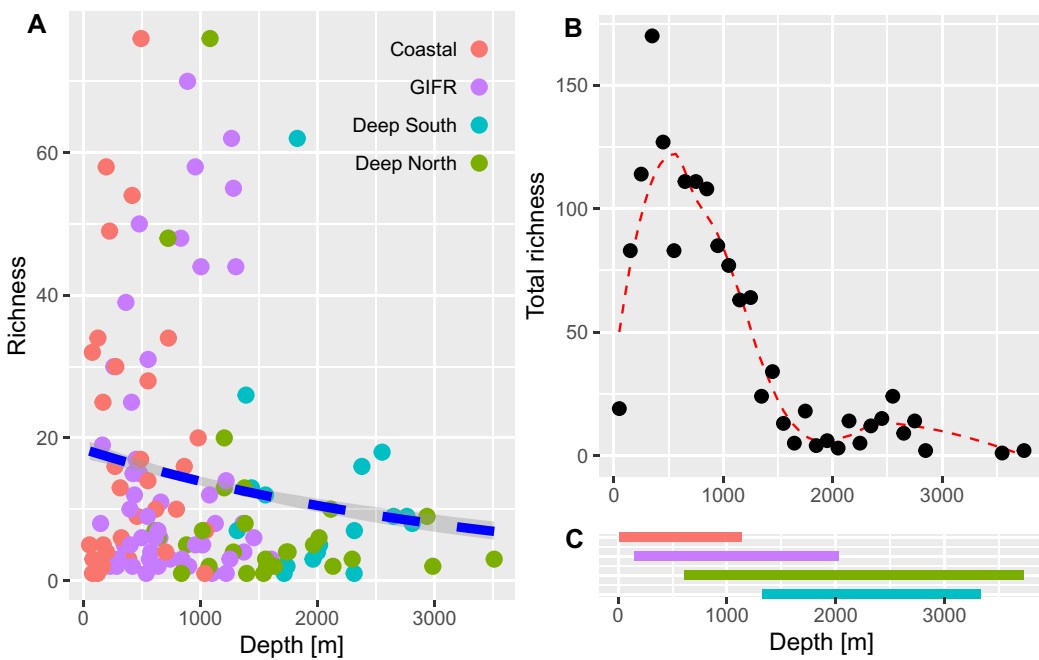

**Figure 5 Amphipod species diversity pattern along a depth gradient.** (A) Richness values per station and coloured according to the environmental clusters. The blue dashed line represents the Poisson GLM. (B) Maximum number of amphipod species per 100-m interval. A Loess smoother represented by the red dashed line is plotted to better visualize the pattern. (C) Bars show the depth ranges in meters for each of the four environmental clusters. Colours according to the legend in (A).

The presence of the GIFR is known as an effective barrier to disrupt the dispersal of benthic organisms between the North Atlantic and the Nordic seas (*Weisshappel & Svavarsson, 1998*; *Brix & Svavarsson, 2010*; *Schnurr et al., 2018*). With a saddle depth averaging 600 m in the Strait of Denmark and 480 m between Iceland and the Faroe Islands and a maximum depth of c. 840 m, the depth increases towards the abyssal basins on each side of the ridge exceeding 3,000 m. Depth, or rather ecological and environmental variables that change with depth, such as hydrostatic pressure, temperature, food availability, or competition, have been demonstrated to have a large impact on species distributions (*Rex & Etter, 2010*; *Brown & Thatje, 2011*; *Tittensor et al., 2011*). In contrast, there are several examples of amphipod species, mostly within the more motile scavenger and predator guilds, with large depth distributions and thus at least the intrinsic capability to overcome topographical barriers (*Lacey et al., 2018*; *Lörz, Jażdżewska & Brandt, 2018*; *Weston et al., 2021*).

The GIFR also marks the transition between different bodies of water, and hence the effects of depth and water mass properties are intertwined. Generally, physical and chemical water mass attributes such as temperature, salinity, pH, organic matter, and dissolved oxygen play critical roles in structuring benthic communities including microbes, fish, crustaceans, corals, and sponges (*Koslow, 1993*; *Weisshappel & Svavarsson, 1998*; *Brix & Svavarsson, 2010*; *Schnurr et al., 2018*; *Puerta et al., 2020*; *Roberts et al., 2021*). Reasons for this involve physiological tolerances of larvae, juveniles and adults towards

certain environmental conditions, dispersal constraints invoked by density differences or current shear, as well as enhanced nutrient input linked to hydrography (*Puerta et al., 2020*; *Roberts et al., 2021*).

Obviously, cold sub-zero temperatures in the Nordic sea basins restrict species distributions, as only few species are pre-adapted to such low temperatures while withstanding high hydrostatic pressures (*Svavarsson, Stromberg & Brattegard, 1993*; *Brown & Thatje, 2011*). This is supported by the fact that many amphipod species in our study prefer moderate conditions, at least in terms of temperature. Initially, however, species originating from the North Atlantic had to overcome the GIFR and enter the Nordic seas against the overflow water from the Denmark Strait and Faroe Bank Channel (*Yasuhara et al., 2008*), the latter being limited to species with broad bathymetric distributions or eurytherm "shallow"-water taxa. The presence of the GIFR is thereby inevitably linked to the opening of the North-east Atlantic about 55 Mya, representing a barrier between the Nordic seas and North Atlantic ever since (*Hjartarson, Erlendsson & Blischke, 2017*). Alternatively, species from the North Pacific had to cross the Bering Strait sill, and experience subsequent trans-Arctic migration (*Hardy et al., 2011*). While the shelf fauna represents a mixture of North Pacific, North Atlantic and to a lesser extent endemic Arctic fauna (*e.g.*, *Svavarsson, Stromberg & Brattegard, 1993*; *Hardy et al., 2011*), large parts of the contemporary deep-sea fauna of the Arctic and Nordic seas likely originate from the North Atlantic (*e.g.*, *Bluhm et al., 2011* and citations therein; *Svavarsson, Stromberg & Brattegard, 1993*).

In our indicator analysis, species were identified based on their predominant affiliation to certain oceanographic conditions. Identifying areas of endemicity, *Arfianti & Costello (2020)* defined our study area as part of a larger region that comprised North American boreal, Arctic and North Pacific areas. Our results, however, are consistent with the view that the deep-sea fauna of the Nordic seas appears to originate from shelf genera or less pronounced deep-sea taxa that were able to cross the GIFR (*Dahl, 1979*; *Just, 1980*; *Svavarsson, Stromberg & Brattegard, 1993*). The study by *Arfianti & Costello (2020)* contained data for the entire Arctic and sub-Arctic regions, encompassing both shelf and deep-sea areas, with the first reportedly representing a mixture of Atlantic, Arctic and Pacific elements (see above). Contrasting distribution patterns in hyperbenthic Eusiridae and Calliopiidae represent good examples to illustrate the barrier effect of the ridge; the family Eusiridae, which is more prevalent in deep water, has only a few species north of the GIFR, which is in contrast to the shallow water family Calliopiidae, whose species diversity is higher in the north (*Weisshappel, 2000*; *Weisshappel, 2001*). Overall, *Svavarsson, Stromberg & Brattegard (1993)* describe the deep-sea fauna of the Arctic and Nordic seas as very young, probably less than 100,000 yrs. old, due to the presence of the ridge and the adverse conditions prevailing in the northern regions ("topographic and environmental filtering"). Accordingly, little time remained for speciation and formation of endemic species (*Svavarsson, Stromberg & Brattegard, 1993*).

Our coastal amphipod assemblage, as well as the one associated with the GIFR, consisted of indicator species with broad North Atlantic distributions. Over the past millennia the biogeography of northern latitudes had been shaped by recurring glacial

cycles (*Darby, Polyak & Bauch, 2006*). During the last glacial maximum (ending about 6,000 yrs ago; *Darby, Polyak & Bauch, 2006*) Arctic shelves were largely covered by grounded ice sheets forcing the fauna towards more southerly (North Atlantic) ice-free areas or deeper waters (*Dunton, 1992*; *Darby, Polyak & Bauch, 2006*). The latter may have become the ancestors of today's Nordic deep-sea fauna (*Nesis, 1984*). While evidence exists that at least parts of the shelf had remained ice-free and thus served as glacial refugia, notably here Iceland and the Faroe Islands (*Maggs et al., 2008*; *Hardy et al., 2011*), most species must have recolonized the previously ice-covered areas rather swiftly. Given the close overlap of coastal and GIFR fauna in our study, the ridge could have provided a potential shallow-water link for brooding taxa that has promoted the recolonization from suitable ice-free habitats.

## Diversity trends

The comparison of the diversity between the environmental clusters showed that the diversity of the shallow clusters (coastal and GIFR) was higher than that of the deep clusters north and south of the ridge. While species richness had the highest number of effective species (Fig. 4A), its sole use is usually not encouraged as it is heavily affected by sample size and shows high sensitivity in recording rare species (*Jost, 2006*). There were some profound differences between Hill numbers—species richness, Shannon, and Simpson diversity—likely because each of these indices scales rarity differently (*Chao, Chiu & Jost, 2014*; *Roswell, Dushoff & Winfree, 2021*; Figs. 4B, 4C). The fact that none of the richness-based rarefaction curves has stabilized yet, could therefore be an artifact; many species have only been found once, either because they could not be identified to species level or because only a small number of individuals were sampled during the historical missions. The Simpson index, on the other hand, is considered as being most robust when sampling effort differs strongly between samples, since it largely reflects patterns in the most common species (*Jost, 2006*). Shannon diversity can be seen as a intermediate measure in terms of its responses to sample size and rarity (*Roswell, Dushoff & Winfree, 2021*). Overall, though, all estimates applied have their merits and pitfalls, and typically using all three indices provides the best representation of the diversity in a given area (*Roswell, Dushoff & Winfree, 2021*). Nevertheless, a consistent pattern of a higher diversity in the shallows—relative to the deep clusters—was evident in all three indices. In the same way, analysis of the entire data set showed an unimodal pattern, with richness peaking at around 500 m, and then a sharp decline in richness with increasing depth (Fig. 5). Compared to other studies that often show a peak between 2,000–3,000 m (cf. *Rex & Etter, 2010* and citations therein), maximum richness in amphipods of the Nordic Seas seems to be much shallower and to resemble patterns in isopods from the same area (*Brix et al., 2018*, but see *Svavarsson, 1997*). However, it should be noted here that differences in sampling intensity between grid cells and depth were a confounding factor in our study and the results therefore will have to be reassessed with additional future sampling.

Combined historical and ecological explanations have been utilized to interpret the overall low diversity of the Nordic basins compared to the other deep-sea regions

(*Svavarsson, Stromberg & Brattegard, 1993*; *Bluhm et al., 2011*). In general, it is believed that variation in energy supply (temperature and productivity) affect deep-sea diversity (*e.g.*, *Woolley et al., 2016*; *Yasuhara & Danovaro, 2016*; *Jöst et al., 2019*). However, cold temperatures *per se* do not seem to have a negative impact on diversity, since benthic communities at sub-zero temperatures in the Southern Ocean abyss appear to be extraordinarily rich (*Brandt et al., 2007*), but when coupled with the very low productivity and geographical isolation of the Nordic basins, the diversity of invertebrates is relatively low (*Svavarsson, Stromberg & Brattegard, 1993*; *Egilsdottir, McGinty & Gudmundsson, 2019*; *Jöst et al., 2019*). In addition, antagonistic effects of high hydrostatic pressure and low temperatures that prevail in the deep Nordic Sea basins could explain the low diversity there (*Brown & Thatje, 2011*, *2014*).

Notably, the diversity of the "Deep South" cluster in our study was as low as that of the Deep North, which contrasts with the perception of an impoverished Nordic deep-sea fauna (*Bouchet & Warén, 1979*; *Dahl, 1979*; *Rex et al., 1993*; *Svavarsson, 1997*; *Weisshappel & Svavarsson, 1998*; *Jöst et al., 2019*). Although amphipods are typically less well presented in the deep sea (*e.g.*, when compared to isopods; *Lörz, Kaiser & Bowden, 2013*), their 'deficiency' in Nordic waters was established earlier. For example, *Dahl (1979)* found that gammaridean species in the Norwegian Sea is a mere 20% of that in the North Atlantic. Yet, it is not clear whether this is a valid conclusion, since pure richness comparisons are very susceptible to differences in sample sizes and sample effort (see discussion above). In addition, different taxa north and south of the ridge can have different diversity patterns resulting *e.g.*, from their different evolutionary histories, lifestyles (brooding *vs.* broadcaster) or physiological scope. This becomes very evident in isopods, a sister group of the amphipods, where the diversity of the deep North Atlantic exceeds that of the Nordic seas (*Svavarsson, 1997*).

Although not strictly comparable, but in line with our results, *Egilsdottir, McGinty & Gudmundsson (2019)*, found local deep-sea diversity of bivalve and gastropod molluscs north and south of the GIFR to be equally low. They attributed this to specific oceanographic conditions prevailing at the deep southern stations. In addition, changes in environmental conditions in the course of past glacial maxima in the northern North Atlantic and in the North Sea were associated with cyclical changes of low (glacial) and relatively increased (interglacial) diversity (*Cronin & Raymo, 1997*; *Yasuhara et al., 2014*). The related environmental consequences of these climatic changes, in particular variation in bottom-water temperature, seasonality and meltwater runoff, evidently had a strong impact on deep-sea diversity, with recent deep-sea fauna still in the process of recovering from these events (*Rex et al., 1993*; *Cronin & Raymo, 1997*; *Wilson, 1998*; *Yasuhara et al., 2008*; *Yasuhara et al., 2014*; but see *Jöst et al., 2019* and citations therein).

Compared to the deep-sea cluster, the diversity of the shallower coastal and GIFR clusters was considerably higher (Fig. 4A). This is in stark contrast to an allegedly poor amphipod fauna, for example when compared to the South polar region (*Arfianti & Costello, 2020*). Although a direct comparison with other regions at complementary depth is still pending, it is already clear that the shelf and upper slope amphipod fauna on the border between the North Atlantic and North Sea, consisting of more than 300

effective species, is not depleted (Fig. 4A). In comparison, *De Broyer & Jazdzewska (2014)* counted ~560 amphipod species for the entire Antarctic region (south of the Polar front), which is considered to have a significantly higher amphipod diversity relative to high northern latitudes (*Arfianti & Costello, 2020*). In addition, through the application of molecular techniques, but also additional sampling, especially of the deeper and less frequently explored areas, more species are likely to be discovered for the northern region (*Bluhm et al., 2011*; *Jażdżewska et al., 2018*; *Lörz, Jażdżewska & Brandt, 2018*; *Schwentner & Lörz, 2020*). We admit the comparison is slightly misleading, as cryptic species are discovered across all environments at similar rates (*Pfenninger & Schwenk, 2007*), plus different geological histories, oceanographic settings, and the size of the Arctic *vs.* Antarctica, among other things represent additional confounders. We thus believe that the diversity of the northern regions should not be underestimated and presumably occupies globally at least a middle ranking.

## CONCLUSIONS

In amphipods, water mass properties appear to be the main force in delineating species distributions at the boundary between the North Atlantic and the Nordic seas, with the GIFR additionally hindering the exchange of deep-sea species between northern and southern deep-sea basins. This pattern is largely congruent for all benthic but also hyperbenthic amphipod families. Different factors are likely responsible for driving deep-sea diversity on each side of the ridge. While impoverished amphipod communities in the Nordic basins are likely to be due to topographical and environmental barrier effects, the southern deep-sea assemblage shows similarly low diversity, presumably a response to variation in the oceanographic environment over a range of temporal and spatial scales. In addition, bathymetric sampling constraints need to be considered.

Since the Cenozoic Era (c. 65 mya) and more recently, the areas of the northern North Atlantic and the Nordic seas have undergone profound climatic changes, from greenhouse to icehouse conditions and vice versa, shaping the composition and distribution of the marine biota (*Piepenburg, 2005*; *Horton et al., 2020*). Distinct temperature thresholds for the Arctic and boreal benthic species point towards future range shifts (restrictions *vs.* extensions), which will have a strong impact on the diversity in the region (*Renaud et al., 2015*). Our data showed a high salinity and temperature-driven distribution of the amphipod assemblages, which also applies to a number of other taxa (*Brix & Svavarsson, 2010*; *Schnurr et al., 2018*; *Egilsdottir, McGinty & Gudmundsson, 2019*; *Jöst et al., 2019*). Additional environmental variables may prove important in explaining diversity and distribution, including seasonality in productivity, pH and ice cover (*Yasuhara et al., 2012*). These are especially the ones that are predicted to change first due to recent climate changes (*e.g.*, *Hoegh-Guldberg & Bruno, 2010*).

In our study, amphipods were highlighted as an important benthic component in Icelandic waters. Since climate change is supposed to have an impact on several organizational levels (populations, species, communities), in future studies, we aim to investigate the interaction of local and regional processes on amphipod diversity as well as

species-specific responses to better understand potential effects of climate change in the Nordic seas.

## ACKNOWLEDGEMENTS

We submitted the data to GFBIO and OBIS (Oceanographic Biodiversity Information System) and are grateful for their processing. We thank all crew and scientific teams for all efforts taking, sorting and identifying the samples. Furthermore, we acknowledge the data management of the IceAGE Amphipoda in the local DZMB database by Antje Fischer and Karen Jeskulke during two amphipod determination workshops in 2016 and 2017.

### Funding

Anne-Nina Lörz was financed by the German Science Foundation project IceAGE Amphipoda (LO2543/1-1). Stefanie Kaiser received a grant from the Narodowa Agencja Wymiany Akademickiej (NAWA, Poland) under the ULAM program. Financial support for two amphipod determination workshops from IceAGE expeditions was given by the Volkswagenstiftung to Saskia Brix and Anne-Nina Lörz. The funders had no role in study design, data collection and analysis, decision to publish, or preparation of the manuscript.

### Grant Disclosures

The following grant information was disclosed by the authors:
German Science Foundation project IceAGE Amphipoda: LO2543/1-1.
Narodowa Agencja Wymiany Akademickiej (NAWA, Poland).
ULAM.

### Competing Interests

The authors declare that they have no competing interests. Jens Oldeland is a freelancing environmental data scientist.

### Author Contributions

- Anne-Nina Lörz conceived and designed the experiments, analyzed the data, authored or reviewed drafts of the paper, and approved the final draft.
- Stefanie Kaiser analyzed the data, authored or reviewed drafts of the paper, and approved the final draft.
- Jens Oldeland performed the experiments, analyzed the data, prepared figures and/or tables, authored or reviewed drafts of the paper, and approved the final draft.
- Caroline Stolter performed the experiments, prepared figures and/or tables, and approved the final draft.
- Karlotta Kürzel performed the experiments, prepared figures and/or tables, and approved the final draft.
- Saskia Brix conceived and designed the experiments, authored or reviewed drafts of the paper, and approved the final draft.

## Data Availability

The data are available in the Supplemental Files.

The data is also available at Pangaea: Lörz, Anne-Nina; Brix, Saskia; Oldeland, Jens; Coleman, Charles Oliver; Peart, Rachel; Hughes, Lauren; Andres, Hans Georg; Kaiser, Stefanie; Stolter, Caroline; Kürzel, Karlotta; Vader, Wim; Tandberg, Anne Helene; Stransky, Bente; Svavarsson, Jörundur; Guerra-García, José-Manuel; Krapp-Schickel, Traudl (2021): Marine Amphipoda and environmental occurrence around Iceland. https://doi.pangaea.de/10.1594/PANGAEA.931959.

## Supplemental Information

Supplemental information for this article can be found online at http://dx.doi.org/10.7717/peerj.11898#supplemental-information.

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
