# Peer review of "Biogeography, diversity and environmental relationships of shelf and deep-sea benthic Amphipoda around Iceland"

_PeerJ, doi:10.7717/peerj.11898_

## Round 0.1 · original submission · Major Revisions

Thank you for an interesting paper. I got 3 referees reports which are constructive. In addition, I highly recommended using ES50 (Hurlberts index), simple species number, and/or Hill's numbers (papers by Anne Chao, iNEXT package) rather than the rather meaningless Shannon index. The problem with indices like Shannons and related relative abundance measures is that they are influenced by both abundance and richness. So they can only be interpreted by reporting richness and relative abundance (which then makes the Shannon index redundant). Plus the Shannon index scale is not comparable across studies where measures like the number of species and total abundance are.

Reviewer 1 ·

Basic reporting

The authors present an interesting and comprehensive dataset of peracarid crustaceans off Iceland covering an extended bathymetric range from continental shelf to abyssal depths. The analysis is based on high taxonomic resolution which is rare for any such macroecological work and therefor of great value. The English language use is very good, the manuscript is well structured and easy to follow, and the figures are clear and follow the methodological description. I do endorse publication of this work following some revisions, the nature of which I have described further below. I congratulate the authors on this interesting piece and am looking forward to reading a thoroughly revised version, once published.

Experimental design

The methodological approach is sound and well justified.

Validity of the findings

The findings are useful and valid within a local as well as global marine biodiversity context.

Additional comments

Abstract and throughout manuscript:
Define better what you mean by climate change effects on the area under investigation, and how these effects may alter/shift diversity and assemblage distribution across the 4 zones (conclusions/outlook).

Can the 4 clusters/zones be characterised by predominant peracarid families/genera or similar? In traditional assemblage analysis, it is common to name species assemblages by for example choosing numerically abundant taxa to name these. These could represent taxonomic state, ecology (eg feeding or specific reproductive mode) that help the reader associating an association with habitat type or ecological niches, and similar.

Why is there no overall representation of diversity – depth pattern irrespective of the 4 zones? I would suggest plotting this for the overall dataset, which then will be eluded upon by discussing the environmental variables for each zone and attempting to identifying the factors that may govern the observed overall pattern the most (similar to what is done for the 4 zones in the present manuscript). Also, this will allow the authors to compare overall diversity -depth pattern with those from other parts of the ocean, in particular for the North Atlantic (eg. Carney, R. S. (2005). Zonation of deep biota on continental margins. Oceanography and Marine Biology: An Annual Review 43, 211–278.), and whether the Icelandic patterns conform to other observations and given that eg temperature changes with depth are less pronounced than at lower latitudes. Incorporate this aspect throughout the manuscript, please. I believe this – though generalising aspect- makes the paper more appealing to macroecologist/physiologists and a wider audience rather than those only interested in peracarids and ‘Icelandic’ diversity.

Results section:
The environmental variables characterising the 4 zones should be better defined/described; something like “low temperature” or “higher pH and dissolved oxygen” is too cryptic. Prvode means …ranges and similar and supporting references.

Discussion:
Effects of depth and hydrostatic pressure. The two citations provided are valid, but the authors may wish to add to the discussion by further referring to a review in Biol. Rev. (see below) that helps identifying physiological factors that govern depth diversity patterns. Depth alone has traditionally been used as a variable, but it indeed is hydrostatic pressure rather than depth as a location proxy that determines diversity patterns. The text from line 351 on will benefit from this discussion of pattern-determining diversity with depth/increase in hydrostatic pressure.
Please do consider the following review within this context:
Brown, A., S. Thatje (2014). Explaining bathymetric diversity patterns in marine benthic invertebrates and demersal fishes: physiological contributions to adaptation of life at depth. Biological Reviews, 89: 406–426.

Line 499: split “webelieve”

Reviewer 2 ·

Basic reporting

The authors clearly stated the purpose and problems to solve in the manuscript. The title reflects the contents of the paper. The abstract describes the vital information in the work, and the introductory section sufficiently explains the framework and problems of the research. However, I found some obvious errors, and I have made several comments on the abstract and introduction sections as below:
Line 49. How many records? The dataset includes genus-level data. Thus, I think it is not 357 species.
Line 53. Why only salinity and temperature? How about depth? As the authors mentioned that the most important environmental variables were temperature, depth, and salinity (line 276-278).
Line 55. Is it GIF or GIFR?
Line 65. The same word was used in one sentence. Suggest revising.
Line 96. Please check the spelling for the cited literature.
Line 121-124. Missing citation.
Line 128. Can be more specific, which literature?
Line 129. How many records?

Experimental design

This is a well-defined research question. However, I have some notes on the methods section as below:
iNEXT package user guide (page 2 & 5) has shown that the package allows incidence data to perform analyses if abundance data is not available or incomplete. The authors indicated that they did not have complete data on abundance (line 143-145). Please explain why the authors used abundance data.
Line 139. How many occurrences? Where is abundance data?
The dataset includes genus-level data; thus, it needs to be clarified how many species and genus.
Line 140-141. This sentence can be omitted as a bit confusing. It will be more concise if the authors refer to a table that contains such information.
Line 141. Can be more detail on “extensive literature search.”
Line 143. Please be more specific, how many?
Line 186. I could not find Table or Figure S3.
Check the requirement for the first mention for an acronym.
Line 198. Check the requirement for the first mention for an acronym.
Line 217. Can be more specific, what is being compared? Which literature? What information?
Line 218. Check the requirement for the first mention for an acronym.

Validity of the findings

The data seem robust, although there are some points that need to be clarified as below:
Table 1. Is it Vel or Velo? Please be consistent.
Table 2. Suggest revising Nr. to No and check the writings in the “species” column.
Some genus levels are still found in this table and were not given a numerical code as indicated by the authors in line 245.
Figure 1. Why are some lines in hexagons bold?
Figure 2. I could not find the appendix file that the author refers to in the figure.
Suggest removing vertical writings on the x-axes as legend has been provided
Figure 3. Missing legend, which one is Fig. 3A and B?
Figure 4. The Shannon-wiener index value increases with biodiversity. Conversely, for the Simpson index, a higher value indicating high dominance or low biodiversity. Please explain why in Fig. 4C coastal cluster has the highest Simpson value.
Supplemental tables. I would suggest combining those two files into a table and add information on the abundance data dan reference. The dataset is not massive, and I think it is worth moving it to the main section.
Line 244. How many occurrences data?
Line 245. There is no numerical code after the genus name in the dataset, such as in row 19. It only written Ampelisca Krøyer, 1842.
Line 271-289. How about other environmental parameters were shown in Fig. 3? Why no mention of other environmental parameters such as phybio and velo?
Line 318. How many individuals?
Line 330. The authors’ claim that the study was built on an extensive data set is a little bit excessive considering that the authors did not provide complete data on the occurrences and abundance and only employed 357 amphipods (species and genus).
Line 334-336. Why is the sentence about comparing diversity in the Nordic basins and the deep-sea coming after the authors’ statement that GIFR had a strong overlap with a coastal assemblage? Suggest adding a sentence that explains the cause of the overlap.
Line 364. Suggest revising the “incl.”
Line 403-414. This paragraph contains unfinished sentence, such as in line 406 and the extensive discussion part almost obscuring the main messages.
Line 416-500. The main point that I get from the “Diversity trends” section is that the diversity of the shallow clusters was higher than the deep clusters, which is valid considering the results presented in the manuscript. However, it could be written in a much shorter section. The section also needs to be organized to not obscure the main findings (idea) by supporting information.
Line 499. Missing space.
Line 518. Missing the article, please check other places too.

Additional comments

I think this is a relevant manuscript, but I feel that the discussion section should be simplified to make the main point clearer. Please complete the data and include all necessary information.

·

Basic reporting

The manuscript is well written in clear unambiguous English, and clearly structured. I have made a few minor suggestions to improve this on the attached PDF. Literature is sufficiently cited throughout, although reference to the World Amphipoda Database is not cited despite being used and therefore needs to be included. It is also necessary for the data tables and the dataset in OBIS to be cited in the paper.

Experimental design

The research aims are clear and relevant. The methods are well-described and in sufficient detail.

Validity of the findings

The findings of the research are interesting and clearly stated and discussed.

Additional comments

This is a well-written and clearly structured paper and should be published subject to the minor revisions suggested.

---

## Round 0.2 · Minor Revisions

Thank you for a good paper. In addition to the referee's constructive recommendations: please avoid missing parentheses -- a) should be (a);
where the text says biogeographic I think it means geographic - all the study area is in one biogeographic region and all species could have colonised all the study area. thus their distributions as shown are defined by the environment, not biogeographic history;
text on graphs should be larger (fig4), black (not grey) and grey backgrounds removed to increase contrast;

Reviewer 4 ·

Basic reporting

The manuscript deals with pattens of biogeography and diversity in a remarkable region. The manuscript is well written and constructed and should be published with minor revision.
The manuscript holds all of the relevant references. The background is clear and the questions asked are highly relevant.
I find the text on the older amphipod data could be extended. Much of the earlier data comes probably from the extensive Ingolf expedition, which was one of the major expeditions to the region at that time. Stephensen published a series of paper on the amphipods from the Ingolf expedition, but only a single of his paper is cited (but probably used in the analysis) dealing with material from the Ingolf. In a way this is a high-quality data. A fine mesh net to get small crustaceans was probably used for the first time in this expedition.
Figure 5 can be improved substantially. First, there are no labelling of A, B and C in the figure. The figure would benefit from having icons (dot, box, triangle, etc.) instead of differently colored dots (which are un-distinguishable on the print-out). The Deep South and the Deep North seem to differ in the left panel, but to be similar in the right panel. Distinguishing between the DS and DN would be beneficial in the right panel.
Line 86. Are the limits of the deep-sea at 150 m?
Lines 102-103. The ridge “… hinders direct flow …”. This is a strange statement.

Experimental design

The environmental layers should be presented more thoroughly. Are these layers near-bottom layers or shallow/midwater layers? Is the resolution within the layers similar in all parts of the region?
Also, some the layers should be termed somewhat differently. While temperature is without a doubt an environmental parameter felt by any species, I guess phytobiomass is hardly environmental parameter, but more of a community descriptor. But I feel that all of these should be used in the analysis.
The authors state that the mclust algorithm identified six clusters. It would be worthwhile showing these clusters in a figure, as it seems there is something interesting going on here, even though there are disproportionally large differences.

Validity of the findings

This is a solid work and very interesting results are coming out of the study. I can highly recommend its publication in PeerJ.

---

## Round 0.3 · accepted · Accept

Thank you for the revisions which have well addressed all referees and my suggestions.